# GRAPH-NEURAL TANGENT KERNEL/GAUSSIAN PROCESS FOR NODE CLASSIFICATION/REGRESSION

## ABSTRACT

This work analyzes Graph Neural Networks, a generalization of Fully-Connected Deep Neural Nets on graph structured data, when their width, that is the number of nodes in each fully-connected layer is increasing to infinity. Infinite Width Neural Networks are connecting Deep Learning to **Gaussian Processes** and **Kernels**, both Machine Learning Frameworks with well-established theoretical foundations. Gaussian Processes and Kernels have less hyperparameters than Neural Networks and can be used for uncertainty estimation, making them more user-friendly for applications. This works extends the increasing amount of research connecting Gaussian Processes and Kernels to Neural Networks. The Kernel and Gaussian Process closed forms are derived for a variety of architectures, namely the standard Graph Neural Network, the Graph Neural Network with Skip-Concatenate Connections and the Graph Attention Neural Network. All architectures are evaluated on a variety of datasets on the task of transductive Node Regression and Classification. Additionally, a Spectral Sparsification method known as *Effective Resistance* is used to improve runtime and memory requirements. Extending the setting to inductive graph learning tasks (Graph Regression/ Classification) is straightforward and is briefly discussed in 8.1.

## 1 INTRODUCTION

Graph Neural Networks (GNNs), introduced in the paper (Kipf and Welling, 2017) demonstrated their effectiveness in tasks involving graph-structured data. GNNs have become a staple tool for data scientists and a prominent area of research in Machine Learning. In recent research, GNNs have been extended to different architectures such as Graph Attention Neural Networks (Veličković et al., 2018), Transformer Graph Neural Networks (Shi et al., 2021) and to unsupervised learning tasks with a Graph (Variational) Autoencoder (Kipf and Welling, 2016). Another very active area of research are infinite width Neural Networks, also called Neural Tangent Kernels (NTK), introduced in the paper (Jacot et al., 2018). NTK link infinite width Fully-Connected Deep Neural Nets (FCN) and Kernels. NTK theory has been expanded to various architectures (Yang, 2020) and used to analyze theoretical properties like Generalization and Optimization (Arora et al., 2019a; Du et al., 2019a) and the importance of appropriate random initialization for successful training (Xiao et al., 2020; Seleznova and Kutyniok, 2022). The NTK Theory has been expanded to address inductive Graph Learning Tasks in Du et al. (2019b). In the Related Work Section, we will compare their results and their relevance to our approach. This work extends infinite width Neural Network theory to the setting of Graph Neural Networks. The main contributions of this work[1] are the derivation of closed-form expressions for the Gaussian Process (GNNGP) and Graph Neural Tangent Kernel (GNTK) for three different architectures: vanilla GNN, Skip-Concatenate GNN, and Graph Attention Neural Network applied to the task of Node Regression/Classification. In addition, we extend the understanding of the GNTK formula's

---

[1]Code is available at Github

positive definiteness by exploring the impact of different graph adjacency matrices, as demonstrated in 3.2 and Section 8.2. Furthermore, we include the evaluation of GNNGP and GNTK against their Neural Network counterparts across various datasets. Additionally, we explore the application of *Effective Resistance* as a Spectral Graph Sparsification Method, aiming to enhance memory and runtime efficiency. Note that the GNNGP for vanilla GNN and a variant of our Skip-GNN has previously been established by (Niu et al., 2023).

## 1.1 NOTATION

Superscript for matrices (i.e. $W^L$) are identifiers and not matrix powers. $I_n \in \mathbb{R}^{n \times n}$ is the identity matrix. We define the operator batchmultiply; $\mathrm{batchm}(X, Y)_{ij} = \langle X, Y_{IJ} \rangle_F$ with $X \in \mathbb{R}^{n \times n}$ and $Y \in \mathbb{R}^{zn \times zn}$ for some integer $z$. So $\mathrm{batchm}(\cdot, \cdot)$ is taking the Frobenius Inner Product of $X$ and block $IJ$ of size $n \times n$ of $Y$ and $\mathrm{batchm}(X, Y) \in \mathbb{R}^{z \times z}$. Concatenating vectors/matrices horizontally is denoted $\mathrm{concat}(\cdot, \cdot)$. The Hadamard Product is denoted with $\odot$. Convergence in probability is denoted as $\xrightarrow{P}$.

## 1.2 GRAPH NEURAL NETWORKS GENERALIZE FULLY-CONNECTED DEEP NETS

GNNs are a powerful generalization of Fully-Connected Nets (FCN). GNNs are extending the FCNs by making use of dependent data samples, naturally incorporating unlabeled data, referred to as semi-supervised learning and allowing for transductive learning tasks such as Node Regression and Classification as well as inductive tasks like Graph Regression/ Classification. We will start by establishing this connection between FCNs and GNNs. A three layer FCN can be defined as follows:

$$f(x) = W^3 \sigma(W^2(\sigma(W^1 x)) \in \mathbb{R}^{d_3} \text{ with } x \in \mathbb{R}^{d_0} \text{ and weights } W^l \in \mathbb{R}^{d_l \times d_{l-1}}. \tag{1}$$

The single data point $x$ is a column vector of a data matrix $X \in \mathbb{R}^{d_0 \times n}$. Now defining a GNN, as in Kipf and Welling (2017). Given a graph adjacency matrix $A \in \mathbb{R}^{n \times n}$ and the data $X^T \in \mathbb{R}^{n \times d_0}$, the GNN is:

$$g(X) = A\sigma(A\sigma(AX^T W^1)W^2)W^3 \in \mathbb{R}^{n \times d_L} \tag{2}$$

with $W^l \in \mathbb{R}^{d_{l-1} \times d_l}$. There are different options for the adjacency matrix $A$. The adjacency matrix can be the 0-1 adjacency matrix of an undirected graph, denoted as $A_{0\text{-}1}$. $A$ can be $A_{0\text{-}1} + I$, which is the 0-1 adjacency matrix with self loops. $A$ can be the Laplacian $L = D - A$ with $D$ being the degree matrix, $D_{ii} = \sum_{ij} A_{ij}$. One can use row, column or row & column normalization on the Laplacian. $A$ can be Degree normalized according to Kipf and Welling (2017) which is $A = (D+I)^{-\frac{1}{2}}(A+I)(D+I)^{-\frac{1}{2}}$, with D being defined as above. If we set $A$ as the Identity matrix $I$ and take the transpose of $g(X)$, we end up with:

$$g(X)^T = W^{3^T} \sigma(W^{2^T}(\sigma(W^{1^T} X^T)) \in \mathbb{R}^{d_3 \times n} \tag{3}$$

$g(X)^T$ is equivalent to a FCN $f(X)$ where we pass as input all the (training and test) data $X$ instead of a single data point $x$. In the case of Node Regression, the training loss is minimized as follows:

$$\underset{W}{\mathrm{argmin}}(||(Y^T - g(X)^T)I_{train}||_F^2) \text{ with } I_{train} = \mathrm{diag}([0, 0, 1, 0, 1, ..., 0, 1]) \tag{4}$$

$I_{train} \in \mathbb{R}^{n \times n}$ consists of randomly selected diagonal 0,1 entries which correspond to the training samples. $I_{test}$ is defined as $I_{test} = I_n - I_{train}$. We have labels $Y \in \mathbb{R}^n$ and $I_{train}$ is randomly selecting columns of $g(X)^T$ over which the training loss is minimized. Columns in $g(X)^T$ correspond to individual training samples when the GNN reduces to a FCN (i.e. $g(X)^T = f(X)$) or individual nodes in when $g(X)^T$ is a proper GNN, in which case we will call this setup Node Regression/ Classification. The task of Node

Regression/ Classification for the GNN with the Identity matrix is equivalent to training a FCN with Gradient descent. Stochastic Gradient Descent (SGD) can be generalized to GNNs straightforwardly, too.

$$g(X)^T = W^{3^T} \sigma(W^{2^T}(\sigma(W^{1^T} X^T \text{dropout}(I)))) \tag{5}$$

Sampling from $I$ without replacement some subset with size k is equivalent to minimizing our objective using SGD with batch size k. This can be emulated using dropout.

$$g(X)^T = W^{3^T} \sigma(W^{2^T}(\sigma(W^{1^T} X^T \text{dropout}(A))A)A \tag{6}$$

Dropout is commonly applied to both the adjacency matrix $A$ and weights in GNN layers. Recognizing GNNs as generalizations of FCNs, the same applies to GNNGP and NNGP, as well as GNTK and NTK. Substituting the Identity matrix $I$ for $A$ in GNNGP and GNTK reduces to the NNGP and the NTK. Node Regression/Classification with GNNs involves transductive learning, where the model can incorporate test data during training, see (Vapnik, 2006, transductive learning Section 10.1, Setting 1) for a formal definition. The expression $AXW$ in 2 is nothing else $(AXW)_i = \sum_{k \in \text{Neighbors}(i)} x_k W$ (for standard the 0-1 adjacency matrix $A$). This summation includes neighboring nodes that could be part of the test set, making it a transductive learning task. Unlabeled nodes, similar to test nodes, can be used during training, hence Node Regression/Classification is sometimes called semi-supervised learning.

## 2 RELATED WORK

The GNTK formulation for inductive Graph Classification/Regression was originally introduced by Du et al. (2019b). Any Node Classification GNN architecture can be adapted for Graph Classification by adding an aggregation layer, such as Max Pooling or averaging, as the final layer (see Section 8.1). Therefore, the GNTK for Node Classification/ Regression is a fundamental building block for constructing the GNTK for Graph Classification. (Yang et al., 2023, Lemma 2) uses (Du et al., 2019b, BLOCK Operation) to derive the closed form expression for a two layer GNTK. We compared their final closed form expression (Yang et al., 2023, Lemma 2) with our own formula 3.2, revealing that our derivation is more general and provides additional insights. (Yang et al., 2023, B.1, Equation 24) results in all the graph information being encoded implicitly. Using B.1, Equation 24 to calculate the closed form for a two layer GNTK requires multiple steps as shown in (Yang et al., 2023, B.2, Equation 24) which are not necessary in our expression. In contrast, our derivation, using basic tools from Linear Algebra, results in a distinct GNTK expression with explicit adjacency matrix information. This enables further simplification of the formula, resulting in a more accessible and general GNTK formulation. Unlike Du et al. (2019b); Yang et al. (2023), our work includes experiments related to Node Classification/Regression tasks and simulations in Section 8.2 demonstrating that NTK assumptions do not hold when using a 0-1 adjacency but are satisfied when employing the Kipf & Welling normalized adjacency. The GNTK formulas in this work are easily extended to the inductive learning setting (see Section 8.1). Niu et al. (2023) derived the GNNGP expression for the standard GNN with the 0-1 adjacency matrix known as GIN (Xu et al., 2019), the Kipf & Welling normalized adjacency as defined in Section 1 and two variants of GNNS with Skip-Connections, namely the GraphSage (Hamilton et al., 2017) and GCNII (Chen et al., 2020). These Skip-Connections differ from our Skip-Concatenate, as they involve summation, while ours entails concatenation. Work on GNTK for Node Classification/Regression is explored by (Sabanayagam et al., 2022), although their derivation of the GNTK is incorrect.

**Theorem 2.1 (incorrect**, from (Sabanayagam et al., 2022, Theorem 1 (NTK for Vanilla GCN))). *For the vanilla GCN defined in* (2) *(with the difference that S is the Adjacency Matrix and not A), the NTK* $\Theta$ *is given by*

$$\Theta = \left[ \sum_{i=1}^{d+1} \Sigma_i \odot (SS^T)^{\odot(d+1-i)} \odot \left( \odot_{j=i}^{d+1-i} \dot{E}_j \right) \right] \odot \mathop{\mathbb{E}}_{f \sim \mathcal{N}(0, \Sigma_d)} [\dot{\Phi}(f)\dot{\Phi}(f)^T] \tag{7}$$

*Here $\Sigma_i \in \mathbb{R}^{n \times n}$ is the co-variance between nodes of the layer $f_i$, and is given by $\Sigma_1 := SXX^TS^T$, $\Sigma_i := SE_{i-1}S^T$ with $E_i := c_\sigma \underset{f \sim \mathcal{N}(0,\Sigma_i)}{\mathbb{E}}[\sigma(f)\sigma(f)^T]$ and $\dot{E}_i := c_\sigma \underset{f \sim \mathcal{N}(0,\Sigma_i)}{\mathbb{E}}[\dot{\sigma}(f)\dot{\sigma}(f)^T]$. $\Phi(x) \approx \frac{1}{1+e^{-x}}$ (i.e Sigmoid function).*

Replacing $S$ with the Identity matrix in the GNTK formula does not yield the NTK for Fully-Connected Deep Nets. The final result will be a Kernel matrix with all entries zero other than the entries on the diagonal which can not be correct. For comparison the correct GNTK is given in 3.2, while the NTK expression is given in 8.7.3). Sabanayagam et al. (2023) used this wrong GNTK formula in subsequent work to investigate representational properties. The derivation of the Graph Attention NTK and GP relies on results from Hron et al. (2020); Yang (2019), as the NTK and GP for Attention Neural Networks closely relate to the Graph Attention Neural Network.

## 3 GAUSIAN PROCESSES AND NEURAL TANGENT KERNELS

We will derive closed-form expressions for various infinite width Graph Neural Network architectures. In 8.7, we provide an overview of results on infinite width Fully Connected Deep Networks, as our derivations for infinite width Graph Neural Networks will be based on these fully connected ones.

### 3.1 GRAPH NEURAL NETWORKS

We define the Graph Neural Net recursively similar to the FCN in 8.7. The matrix A is commonly the adjacency matrix with $A \in \mathbb{R}^{n \times n}$, but for the GP and NTK no assumptions about A are used.

**Definition 3.1** (Graph Neural Network).

$$F^h(X) = \left( \frac{\sigma_w}{\sqrt{d_{h-1}}} W^h G^{h-1}(X) + \sigma_b B^h \right) A^T \in \mathbb{R}^{d_h \times n} \text{ with } B^h = b^h \otimes \mathbf{1}_n^T \text{ and } b^h \in \mathbb{R}^{d_h} \quad (8)$$

$$G^h(X) = \sigma(F^h(X)) \text{ and } G^0 = X \in \mathbb{R}^{d_0 \times n} \quad (9)$$

*The weights are initialized as $W^h \sim N(0,1)$ and $b^h \sim N(0,1)$.*

*The final network of depth L is $F^L(X) = \left( \frac{\sigma_w}{\sqrt{d_{L-1}}} W^L G^{L-1}(X) + \sigma_b B^L \right) A^T \in \mathbb{R}^{d_L \times n}$*

**Theorem 3.1** (GNN Gaussian Process). *If all the weight dimensions of the hidden layers $d_{L-1}, d_{L-1}, ..., d_1$ of a GNN, $F(X)^L$ successively go to infinity, then $\text{vec}(F(X)^L) \sim GP(0, (A\Lambda^{L-1}A^T) \otimes I_{d_L})$ with*

$$\Lambda^L = \sigma_w^2 \mathbb{E}\left[\sigma(u)\sigma(u^T)\right] + \sigma_b^2 \quad (10)$$

*with $u \sim N(0, A\Lambda^{L-1}A^T)$*

$$\Lambda^0 = \frac{\sigma_w^2}{d_0} X^T X + \sigma_b^2 \quad (11)$$

The proof of Theorem 3.1 is given in 8.8.1. Theorem 3.1 has been derived by Niu et al. (2023). The expression $\mathbb{E}\left[\sigma(u)\sigma(u^T)\right]$ with $u$ being a multivariate Gaussian vector can be calculated in closed form for certain activation functions (see 8.1 for a detailed discussion).

**Theorem 3.2** (GNN NTK).
*If all the weight dimensions of the hidden layers $d_{L-1}, d_{L-1}, ..., d_1$ of a GNN, $F(X)^L$ successively go to infinity, the NTK has a closed form expression $\Theta^L \otimes I_{d_L}$, with*

$$\Theta^L = A \left( \Lambda^{L-1} + \left( \dot{\Lambda}^{L-1} \odot \Theta^{L-1} \right) \right) A^T \text{ with} \quad (12)$$

$$\Lambda^L = \sigma_w^2 \mathbb{E}\left[\sigma(u)\sigma(u^T)\right] + \sigma_b^2 \tag{13}$$

$$\dot{\Lambda}^L = \sigma_w^2 \mathbb{E}\left[\dot{\sigma}(u)\dot{\sigma}(u^T)\right] + \sigma_b^2 \tag{14}$$

*with $u \sim N(0, A\Lambda^{L-1}A^T)$*

$$\Theta^1 = \Lambda^0 = \sigma_w^2 X^T X + \sigma_b^2 \tag{15}$$

The proof of 3.2 is given in 8.8.2. $\Lambda^{L-1} + (\dot{\Lambda}^{L-1} \odot \Theta^{L-1})$ is positive definite (Jacot et al., 2018, A.4) or (Du et al., 2019a, Theorem 3.1). In the context of the GNTK, the NTK's positive definiteness relies on the positive definiteness of $A$. This might explain why using the Kipf & Welling normalized adjacency matrix, with eigenvalues in the range $[0, 2]$ (Kipf and Welling, 2017), aligns with NTK theory (i.e. loss is going to zero for increases width, weights are close to initialization and the empirical NTK stays close to initialization during training, see 8.7 for details and definition of **empirical** NTK), whereas the 0-1 adjacency matrix, usually not positive semidefinite is not consistent with NTK theory. For simulations confirming this fact, see 8.2. The empirical NTK's positive definiteness during training, aligns with key insights about FCNs. If the empirical NTK stays positive definite during training the loss converges to a global optimum, achieving zero training loss (Du et al., 2019a, Lemma 3.3). Additionally, Nguyen et al. (2021) demonstrates that the empirical NTK of wide Neural Networks remains positive definite during training under mild assumptions on the data distribution. The smallest eigenvalue of the Kernel is related to Generalization Bounds (Arora et al., 2019b) and memorization capacity (Montanari and Zhong, 2022) The GNTK formula may provide insights into the significance of different graph adjacency matrices for GNNs. Also note that the expression reduces to the standard NTK formula for FCNs (see 8.7.3) when replacing A with the Identity matrix.

## 3.2 SKIP-CONCATENATE GRAPH NEURAL NETWORK

Skip-Concatenate Graph Neural Networks (Skip-GNNs), inspired from Chen et al. (2019) consist of concatenating the output of the non-linearity and pre-nonlinearity. The first layer is not affected but starting from the second layer, $W^h$ with $h > 1$ has dimensions $W^h \in \mathbb{R}^{d_l \times 2d_{l-1}}$.

**Definition 3.2** (Skip-Concatenate Graph Neural Network)**.**

$$F^h(X) = \left(\frac{\sigma_w}{\sqrt{2d_{h-1}}} W^h G^{h-1}(X) + \sigma_b B^h\right) A^T \text{ for } h > 1 \tag{16}$$

$$F^1(X) = \left(\frac{\sigma_w}{\sqrt{d_1}} W^1 G^0(X) + \sigma_b B^1\right) A^T \text{ with } B^h = b^h \otimes \mathbf{1}_n^T \text{ and } b^h \in \mathbb{R}^{d_h} \tag{17}$$

$$G^h(X) = \begin{pmatrix} \sigma(F^h(X)) \\ F^h(X) \end{pmatrix} \text{, so } G^h(X) \in \mathbb{R}^{2d_l \times n} \text{ and } G^0 = X \in \mathbb{R}^{d_0 \times n} \tag{18}$$

*The weights are initialized as $W^h \sim N(0,1)$ and $b^h \sim N(0,1)$.*

*The final Network of depth L is $F^L(X) = \left(\frac{\sigma_w}{\sqrt{d_{L-1}}} W^L G^{L-1}(X) + \sigma_b B^L\right) A^T \in \mathbb{R}^{d_L \times n}$.*

**Theorem 3.3** (Skip-Concatenate GNN Gaussian Process)**.**
*If all the weight dimensions of the hidden layers $d_{L-1}, d_{L-1}, ..., d_1$ of a Skip-GNN $F(X)^L$ successively go to infinity, then $\text{vec}(F(X)^L) \sim GP(0, (A\Lambda^{L-1}A^T) \otimes I_{d_L})$, with*

$$\Lambda^L = \sigma_w^2 \frac{1}{2}\left(\mathbb{E}\left[\sigma(u)\sigma(u^T)\right] + A\Lambda^{L-1}A^T\right) + \sigma_b^2 \tag{19}$$

*with $u \sim N(0, A\Lambda^{L-1}A^T)$*

$$\Lambda^0 = \frac{\sigma_w^2}{d_0} X^T X + \sigma_b^2 \tag{20}$$

The proof of Theorem 3.3 is given in 8.9.1.

**Theorem 3.4** (Skip-Concatenate GNN NTK). *Having weight dimensions of the hidden layers* $d_{L-1}, d_{L-1}, ..., d_1$ *of a Skip-GNN* $F(X)^L$, *approaching infinity, the NTK is* $\Theta^L \otimes I_{d_l}$

$$\Theta^L = A \left( \Lambda^{L-1} + \left( \dot{\Lambda}^{L-1} \odot \Theta^{L-1} \right) \right) A^T \tag{21}$$

$$\Lambda^L = \sigma_w^2 \frac{1}{2} \left( \mathbb{E} \left[ \sigma(u)\sigma(u^T) \right] + A\Lambda^{L-1}A^T \right) + \sigma_b^2 \tag{22}$$

$$\dot{\Lambda}^L = \sigma_w^2 \frac{1}{2} \left( \mathbb{E} \left[ \dot{\sigma}(u)\dot{\sigma}(u^T) \right] + 1 \right) + \sigma_b^2 \tag{23}$$

*with* $u \sim N(0, A\Lambda^{L-1}A^T)$

$$\Theta^1 = \Lambda^0 = \sigma_w^2 X^T X + \sigma_b^2 \tag{24}$$

The proof of Theorem 3.4 is given in 8.9.2.

### 3.3 GRAPH ATTENTION NEURAL NETWORK

In this section we recap the Graph Attention Model (GAT) Model from Veličković et al. (2018). We will start by defining a single Attention Layer. We have $X \in \mathbb{R}^{d_0 \times n}$, $W \in \mathbb{R}^{d_1 \times d_0}$ and $A \in \mathbb{R}^{n \times n}$. A standard GNN Layer without bias is $GCN(A, X) = \sigma(WXA^T)$. A GAT Layer is $GAT(A, X) = \sigma_2(WX\alpha(c, A, WX)))$. The attention matrix commonly used for the GAT is the standard 0-1 Attention Matrix $A_{0\text{-}1}$ with added self loops, so $A = A_{0\text{-}1} + I$. We compute $\alpha(c, A, WX)$ as follows:

$$H := XW \in \mathbb{R}^{n \times d_{out}} \tag{25}$$

$$M(H, c)_{ij} = \langle c, \text{concat}(H_{\cdot i}, H_{\cdot j}) \rangle = c_1^T H_{\cdot i} + c_2^T H_{\cdot j} \text{ with } c_1 := c_{[1,...,d_{out}]} \text{ and } c_2 := c_{[d_{out},...,2d_{out}]} \tag{26}$$

$$\alpha(c, A, G)_{ij} = \frac{\exp\left(\sigma_1\left(M(H, c)_{ij}\right)\right)}{\sum_{k \in Neighbors(i) \cup \{i\}} \exp(\sigma_1\left(M(H, c)_{ik}\right)} \tag{27}$$

The final step consists of computing the row wise $\mathrm{Softmax}$ only over neighboring nodes of node $i$ and node $i$ itself. The attention weight vector $c \in \mathbb{R}^{2d_0}$ is a learnable parameter. $\sigma_1$ is an elementwise nonlinearity like the LeakyRelu. Multiple GAT Layers' outputs can be stacked vertically for multiheaded attention, either by concatenation or averaging over different heads ($\frac{1}{H} \sum_h GAT_h(A, X)$). The GAT model we are using to derive the GP and NTK is a simplified version. We replace one of the $W$s by a copy $\tilde{W}$, simplifying the layer to $GAT(A, X) = \sigma_2(WX\alpha(c, A, \tilde{W}X))$. This results in $M(H, c)_{ij} = c_1^T(\tilde{W}X)_{\cdot i} + c_2^T(\tilde{W}X)_{\cdot j} = \tilde{c}_1^T X_{\cdot i} + \tilde{c}_2^T X_{\cdot j}$ so we can just disregard $\tilde{W}$ altogether Disregarding $\tilde{W}$ allows us to apply results from Yang (2019) and Hron et al. (2020). Our simplified GAT model (denoted as GAT*) exhibits no apparent performance loss (see Table 2 and 3). We replace $\mathrm{Softmax}$ with an element-wise nonlinearity since there's no closed-form expression for $\mathbb{E}\left[\mathrm{Softmax}(u)\,\mathrm{Softmax}(u^T)\right]$ with $u \sim N(0, \Lambda)$. For the recursive definition of GAT*, we omit the bias term, intending to include it in the final infinite width limit formula without proof (see Corollary 3.1).

**Definition 3.3** (GAT*). *We have two elementwise nonlinearities* $\sigma_1$ *and* $\sigma_1$ *which are polynomially bounded, (i.e.* $\sigma(x) = c + m|x|$ *for some* $c, m \in \mathbb{R}_+$*) and* $c, c_1, c_2$ *are defined as above, i.e* $c^T = \text{concat}(c_1^T, c_2^T)$.

$$G^l = \sigma_2(F^l) \text{ and } G^0 = X \text{ with } X \in \mathbb{R}^{d_0 \times n} \tag{28}$$

$$L_{i,j}^{l,h} = A_{ij} \frac{\sigma_c}{\sqrt{2d_{l-1}}}(c_1^{l,h^T} G_{\cdot i}^{l-1} + c_2^{l,h^T} G_{\cdot j}^{l-1}) \text{ with } c_1^{l,h}, c_2^{l,h} \in \mathbb{R}^{d_{l-1}} \tag{29}$$

$$F^{l,h} = G^{l-1}\sigma_1(L^{l,h}) \tag{30}$$

$$F^l = \frac{\sigma_w}{\sqrt{Hd_{l-1}}}W^l\left[F^{l,1}, F^{l,2}, \cdots, F^{l,H}\right]^T \tag{31}$$

*with $W \in \mathbb{R}^{d_l \times Hd_{l-1}}$. $F^l$ can also be written as $F^l = \frac{\sigma_w}{\sqrt{Hd_{l-1}}}\sum_h^H W^{l,h}F^{l,h}$ and $W^l, c^{l,h} \sim N(0,1)$.*

*The final Network of depth $l$ and heads $H$ is $F^l = \frac{\sigma_w}{\sqrt{Hd_{l-1}}}W^l\left[F^{l,1}, F^{l,2}, \cdots, F^{l,H}\right]^T$*

### 3.3.1 GAUSSIAN PROCESS

**Theorem 3.5** (GAT* GP)**.**
*If $min\{H, d_{l-1}\} \to \infty$ then $F^{l+1}$ (defined as above) is a Gaussian Process with $\mathrm{vec}(F^{l+1}) \sim GP\left(0, \Lambda^l \otimes I_{d_{l+1}}\right)$*

$$\Lambda^l := \mathbb{E}\left[\sigma_2(u)\sigma_2(u^T)\right] \text{ with } u \sim GP\left(0, \mathrm{batchm}\left(\sigma_w^2\Lambda^{l-1}, \psi(\Lambda^{l-1})\right)\right) \tag{32}$$

$$\psi(\Lambda^l) := \mathbb{E}\left[\sigma_1(v)\sigma_1(v^T)\right] \text{ with } v \sim GP\left(0, \sigma_c^2\gamma_A(\Omega)\right) \tag{33}$$

*and $\gamma_A(\Omega) := J_A(\begin{smallmatrix} \Omega & \Omega \\ \Omega & \Omega \end{smallmatrix})J_A^T$ and $J_A = \mathrm{diag}(\mathrm{vec}(A))\,\mathrm{concat}\left((\mathbf{1}_n \otimes I_n), (I_n \otimes \mathbf{1}_n)\right)$*

$$\Lambda^0 = \frac{1}{d_1}X^T X \tag{34}$$

The proof of Theorem 3.5 is given in 8.10.1.

**Theorem 3.6** (GAT* NTK)**.** *The NTK (of a GAT* of depth L) is $\Theta^L \otimes I_{d_l}$*

$$\Lambda^l := \mathbb{E}\left[\sigma_2(v)\sigma_2(v^T)\right]), \quad \dot\Lambda^l := \mathbb{E}\left[\dot\sigma_2(v)\dot\sigma_2(v^T)\right] \text{ with } v \sim GP\left(0, \Lambda^{l-1}\right) \tag{35}$$

$$\dot\psi(\Lambda^l) := \mathbb{E}\left[\dot\sigma_1(v)\dot\sigma_1(v^T)\right] \text{ with } v \sim GP\left(0, \sigma_c^2\gamma_A(\Omega)\right) \tag{36}$$

$$\Theta^1 = \Lambda^0 = \sigma_w^2\frac{1}{d_1}X^T X \tag{37}$$

$$\Theta^L = \mathrm{batchm}\left[\sigma_w^2\Lambda^{L-1}, \psi(\Lambda^{L-1})\right] \tag{38}$$

$$+ \mathrm{batchm}\left[\sigma_w^2\Lambda^{L-1}, \sigma_c^2\left(\gamma_A(\Lambda^{L-1}) \odot \dot\psi(\Lambda^{L-1})\right)\right] \tag{39}$$

$$+ \mathrm{batchm}\left[\sigma_w^2\Lambda^{L-1}, \sigma_c^2\left(\gamma_A(\Theta^{L-1} \odot \dot\Lambda^{L-1}) \odot \dot\psi(\Lambda^{L-1})\right)\right] \tag{40}$$

$$+ \mathrm{batchm}\left[\sigma_w^2\left(\Theta^{L-1} \odot \dot\Lambda^{L-1}\right), \psi(\Lambda^{L-1})\right] \tag{41}$$

The proof of Theorem 3.6 is given in 8.10.2. For the GAT*NTK without $\sigma_1$ being the identity function, see Corollary 8.2.

**Corollary 3.1** (GAT* NTK with bias (without proof))**.**
*To include a bias term for the GAT NTK one can replace every occurrence of $\sigma_w^2\Lambda^{l-1}$ by $\sigma_w^2\Lambda^{l-1} + \sigma_b^2$.*

## 4 EVALUATION

### 4.1 DATASETS

All models were assessed on various transductive Node Classification tasks using datasets such as "Cora", "CiteSeer" and "PubMed" from Yang et al. (2016) and "Wiki" from Yang et al. (2020). These tasks involve accuracy as the performance measure. Additionally, Node Regression Tasks, including "Chameleon",

"Squirrel" and "Crocodile" from the Wikipedia networks dataset (Rozemberczki et al., 2021), were evaluated using the $R^2$-Score. The datasets are split according to (Kipf and Welling, 2017; Veličković et al., 2018), with a fixed train/val/test split to accommodate graphs with incomplete node labels, like Cora. For Regression tasks, the splitting follows Niu et al. (2023) and is also the same split across different experiments.

Table 1: Dataset summary

| Dataset | Task | Nodes | Edges | Features | Classes | Train/Val/Test Ratio |
|---------|------|-------|-------|----------|---------|---------------------|
| Cora | Classification | 2,708 | 10,556 | 1,433 | 7 | 0.05/0.18/0.37 |
| Citeseer | Classification | 3,327 | 9,104 | 3,703 | 6 | 0.04/0.15/0.30 |
| PubMed | Classification | 19,717 | 88,648 | 500 | 3 | 0.003/0.025/0.051 |
| Wiki | Classification | 2,405 | 17,981 | 4,973 | 17 | 0.60/0.20/0.20 |
| Chameleon | Regression | 2,277 | 62,742 | 3,132 | Regression | 0.48/0.32/0.20 |
| Squirrel | Regression | 5,201 | 396,706 | 3,148 | Regression | 0.48/0.32/0.20 |
| Crocodile | Regression | 11,631 | 341,546 | 13,183 | Regression | 0.48/0.32/0.20 |

## 4.2 MODELS

### NEURAL NETS

The models (FCN, GNN, S-GNN, GAT, GAT*) are Neural Networks. All models, except S-GNN, have two layers, as additional depth did not improve performance. S-GNN has three layers due to Skip-Connections taking effect after a minimum of three layers. FCN is a Neural Net trained solely on node features, equivalent to a GNN with $I$ as the adjacency matrix. The models were trained for 300 epochs without early stopping. Features were normalized before training. Adam optimizer with weight decay of 0.0005 was used for all models. GNN and S-GNN had a learning rate of 0.01, while GAT and GAT* used 0.005. Dropout of 0.5 was applied to every layer's output during training. For GAT and GAT*, 0.5 dropout was also applied to the attention adjacency matrix. Neural Nets used Cross-Entropy Loss for classification and mean-squared error for Regression, following hyperparameters from Kipf and Welling (2017); Veličković et al. (2018). Regression for FCN, GNN, and Skip-GCN had a learning rate of 0.1 without weight decay, with batch normalization after each layer except the last one. GAT and GAT* used a learning rate of 0.01 without batch normalization. FCN for Regression had a Sigmoid nonlinearity which improved performance.

### KERNELS AND GAUSSIAN PROCESSES

The Kernels and GP were trained without feature normalization. Grid search for values between 0.001 and 10 on the validation set determined the best regularization parameters for GP (known as Noise Parameter for GP and Kernel Ridge Regression for the Kernels (Rasmussen and Williams, 2006)). Classification tasks set $\sigma_w^2$ and $\sigma_c^2$ to one, and $\sigma_b^2$ to zero. For Regression, $\sigma_b^2$ was set to 0.1. Additional experiments, see 8.6), investigate the role of $\sigma_b^2$ in the performance of regression tasks, demonstrating its crucial influence. For GAT*-GP and GAT*-NTK the nonlinearity $\sigma_1$ was the Identity Function. $\sigma_2$ is the LeakyRelu with a slope of 0.2 for the negative part. In 6, different nonlinearities for GAT*GP and GAT*NTK are compared. Two nonlinearities instead only one can improve the accuracy by 1%-2%.

## 4.3 EXPERIMENTS

Due to the memory and time requirements for some of the graphs for GAT*GP and GAT*NTK, we "sparsified" graphs denoted with # by removing 90% of their edges using Effective Resistance (Spielman and

Srivastava, 2011). For details, see 8.5. The experiments reveal that NTK and GP are competitive with their Neural Net counterparts, even when the Neural Network is not a vanilla Neural Net trained with MSE Loss using SGD and with extensions like dropout turned off. FCNs, for instance, showed a significant performance gap of 6% - 10% compared to their vanilla counterparts, not considering elements like dropout and batch normalization (Arora et al., 2019a). Batch normalization can be integrated into the GP/NTK framework (Yang, 2020). The GNTK appears to be a robust choice, slightly outperforming GNNGP in Regression. In all cases, a GP/NTK achieves matching or even better performance than Neural Net counterparts. The GAT variants perform poorly for Regression tasks, with some even yielding negative $R^2$-Scores for the GP & NTK counterparts. Overall, using Kernels and GP is preferable due to their less sensitivity to hyperparameters, in contrast to GNNs (e.g., number of hidden nodes, see Table 5). An overview of the environment, implementation details, and runtime (in seconds) & memory requirements (in GB) for different models in 8.4. In 8.5, we explore optimization tricks leveraging the homogeneity of activation functions, like ReLU, for GAT*GP and GAT*NTK implementation. Table 7 contains additional experiments on the impact of different levels of Effective Resistance Spectral Graph Sparsification on the GNNGP and GNTK models.

| | Cora | Citeseer | Pubmed | Wiki |
|---|---|---|---|---|
| FCN | 0.61 | 0.59 | 0.73 | 0.72 |
| GNN | 0.81 | 0.71 | 0.79 | 0.66 |
| Skip-GNN | 0.81 | 0.71 | 0.79 | 0.75 |
| GAT | 0.82 | 0.71 | 0.77 | 0.66 |
| GAT* | 0.82 | 0.70 | 0.77 | 0.65 |
| NNGP | 0.60 | 0.62 | 0.73 | **0.81** |
| NTK | 0.58 | 0.62 | 0.72 | **0.81** |
| GNNGP | **0.83** | 0.71 | **0.80** | 0.78 |
| GNTK | **0.83** | **0.72** | 0.79 | 0.79 |
| SGNNGP | **0.83** | 0.71 | **0.80** | 0.78 |
| SGNTK | 0.80 | **0.72** | 0.79 | 0.78 |
| GAT*GP | 0.79 | 0.71 | 0.73# | 0.78 |
| GAT*NTK | 0.79 | 0.71 | 0.73# | 0.77 |

Table 2: Classification

| | Chameleon | Squirrel | Crocodile |
|---|---|---|---|
| FCN | 0.52 | 0.39 | 0.75 |
| GNN | 0.48 | 0.35 | 0.64 |
| Skip-GNN | 0.38 | 0.31 | 0.58 |
| GAT | 0.43 | 0.30 | 0.66 |
| GAT* | 0.54 | 0.30 | 0.65 |
| NNGP | 0.63 | 0.45 | 0.79 |
| NTK | 0.67 | 0.48 | **0.80** |
| GNNGP | 0.64 | 0.48 | 0.78 |
| GNTK | **0.68** | **0.51** | 0.79 |
| SGNNGP | 0.59 | 0.43 | 0.64 |
| SGNTK | 0.57 | 0.46 | 0.55 |
| GAT*GP | -12.14 | -8.6# | 16.94# |
| GAT*NTK | -8.75 | -8.2# | -17.69# |

Table 3: Regression

## 5 DISCUSSION

Developing closed form expression for different GNN architectures is an important step in paving the way to generalize results from NTK Theory for GNNs. As discussed in 3.2, the spectrum of the NTK plays a crucial role and needs to be investigated further. The GAT models are particularly interesting because the adjacency matrices are learned, adding to the complexity. Nevertheless, NTK Theory has to be considered with care as distinctions between GNNs and FCNs arises. GNNs appear more susceptible to overfitting when overparametrized and employing optimization algorithms like Adam, as opposed to GD/SGD, becomes essential for achieving good results (see 4).

## 6 CONCLUSION

We developed novel Kernel and GP models, leveraging their link to infinite width Neural Networks. Demonstrating competitive performance with recent GNN architectures, they offer a viable alternative for graph structured data tasks. These models, being easy to implement, also enable uncertainty estimation. We aim for this work to inspire the creation of innovative Neural Network architectures and Kernel designs.

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

## 8 APPENDIX

### 8.1 INFINITE WIDTH GRAPH NEURAL NETWORKS FOR GRAPH REGRESSION/ CLASSIFICATION

In the setting of inductive learning, the task is to train a GNN on multiple graphs, e.g.:
$((A_1, X_1, y_1), (A_2, X_2, y_2), ..., (A_n, X_n, y_n))$. Where each triple consists of a graph, its features and a label (like a class assignment for classification). A two layer GNN with a final sum pooling layer, can be written down as: $F^3(A_i, X_i)_h = \sum_h F^2(A_i, X_i)_{\cdot h}$. So $F^3(A_i, X_i) \in \mathbb{R}^{d_3}$ is a vector and $F^2(A_i, X_i) \in \mathbb{R}^{d_2 \times d_3}$ is defined like the GNN for Node Regression/ Classification. The infinite GP in that case will be $\text{vec}(F^3) \sim GP(0, \Sigma)$ with $\Sigma_{ij} = \sum_{st} \left(A_i \Lambda^1 A_j^T\right)_{st}$.

$$\Lambda^1 = \sigma_w^2 \mathbb{E}\left[\sigma(u)\sigma(u^T)\right] + \sigma_b^2$$

with $u \sim N(0, A_i \Lambda^0 A_j^T)$

$$\Lambda^0 = \frac{\sigma_w^2}{d_0} X_i^T X_j + \sigma_b^2$$

The GNTK extends similarly to the setting of Graph Induction. The NTK is $\sum_{st} \Theta^L(i, j)_{st}$ with

$$\Theta^L(i, j) = \sum_{st} \left(A_i \left(\Lambda^{L-1} + \left(\dot\Lambda^{L-1} \odot \Theta^{L-1}\right)\right) A_j^T\right)_{st} \text{ with}$$

$$\Lambda^L = \sigma_w^2 \mathbb{E}\left[\sigma(u)\sigma(u^T)\right] + \sigma_b^2$$
$$\dot\Lambda^L = \sigma_w^2 \mathbb{E}\left[\dot\sigma(u)\dot\sigma(u^T)\right] + \sigma_b^2$$

with $u \sim N(0, A_i \Lambda^{L-1} A_j^T)$

$$\Theta^1 = \Lambda^0 = \sigma_w^2 X_i^T X_j + \sigma_b^2$$

$\Theta^L(i, j)$ is just the NTK for Node Classification/ Regression for different data samples $(A_i, X_i)$ and $(A_j, X_j)$. The proof can be conducted using the same approach as in 8.8.

### 8.2 SIMULATIONS

In this section we empirically confirm the NTK findings from 8.7. 1) consists of showing the for increased width the loss converges to zero, 2) consists of showing that the weight barely change (in relative frobenious norm) for increased width and 3) consists of showing that the NTK is close to its initialization during training (in relative frobenious norm). We use the KarateClub dataset (Zachary, 1977) with 34 nodes, of which we used 27 for training with four classes. We now show for FCNs, GNNs, Skip-GNNs and the GAT that 1), 2) and 3) are valid (see Fig. 1). They were initialized randomly as defined in 8.1, 3.1, 3.2 and 3.3. For the GNN and Skip-GNN we use the Kipf & Welling normalized adjacency matrix and for the GAT the 0-1 adjacency with added self loops. Finally, using a GNN with the 0-1 adjacency with self loops is not consistent with NTK theory. Figure 2 shows clearly that with increased width neither the weights are changing less nor does the empirical NTK stay close to its initialization. For a detailed discussion on why, see 3.2.

Table 4 displays the training loss and accuracy comma-seperated for a two layer GNN with increasing width for different datasets. In table 4 the GNN was trained with Gradient Descent for with a learning rate of 0.001 and in table 5 Adam Optimizer with the same learning rate. For a FCN we would expect a very wide Neural Net to achieve zero training loss and **not** overfit (Arora et al., 2019a). For GNNs we can observe that Gradient Descent fails completely, although the loss in each case is close to zero. For the adam optimizer increased width, improves accuracy with an outlier when we set the width to 128. The loss is already near zero for setting the width to 8. GNNs seem to be very sensitive to the optimizer, step size and width more than would be expected for FCNs.

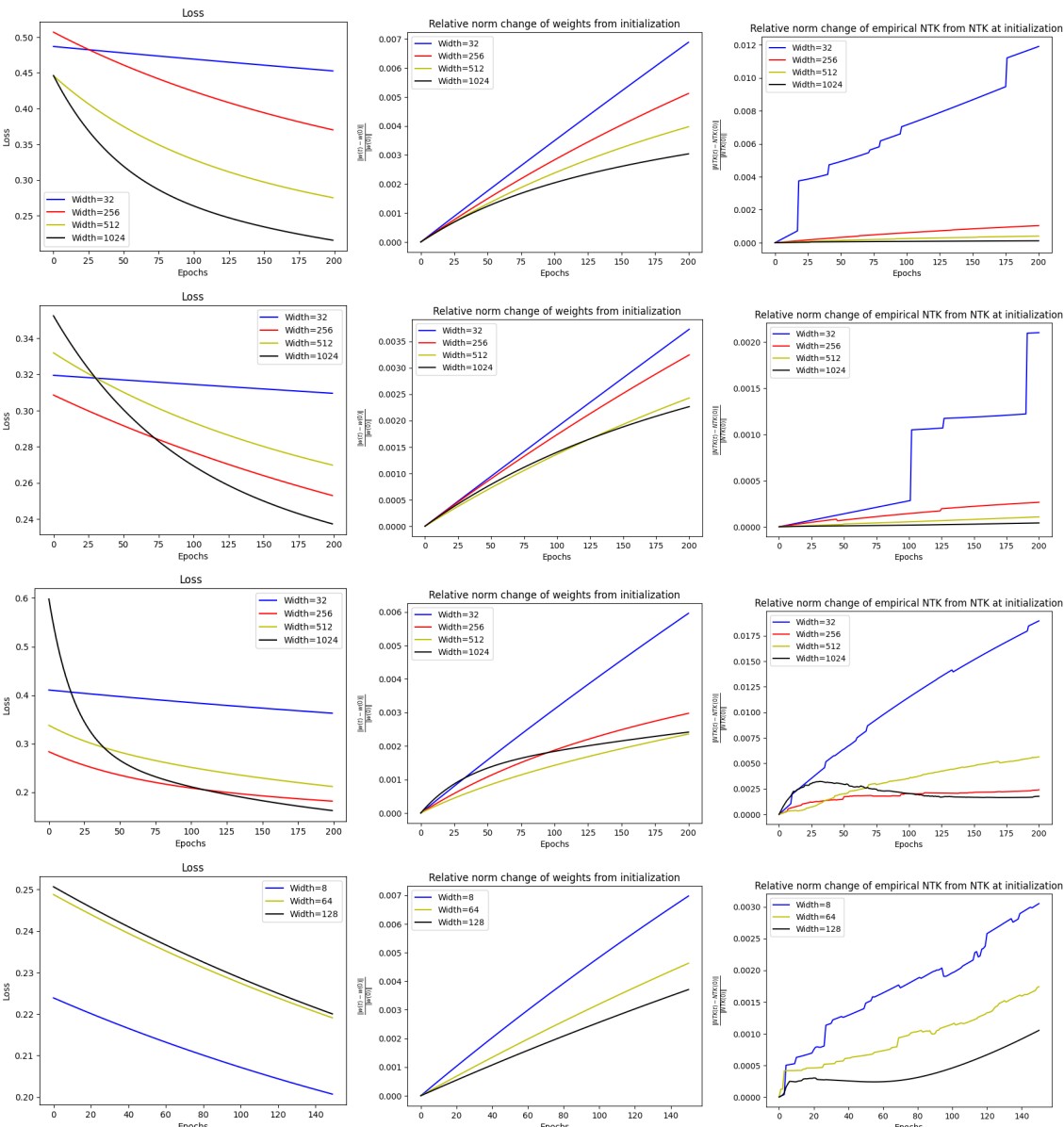

Figure 1: First row corresponds to a two layer FCN, second row to a two layer GNN with the Kipf & normalized adjacency matrix, third row is the Skip-GNN with the same adjacency matrix and fourth row a GAT with the number of heads equal to the width and the attention heads are summed over. All models are trained with learning rate of 0.001 and Gradient Descent. For each row the left figures shows training loss for different widths, the middle figure shows weight change (difference to weights at initialization) during training and the right figures show the difference of the empirical NTK compared to its initialization during training. These simulations confirm that for increased width during training we can observe 1) the training loss approaches zero 2) the weights of the GNN stay close to its initialization and 3) the empirical NTK stays close to its initialization during training.

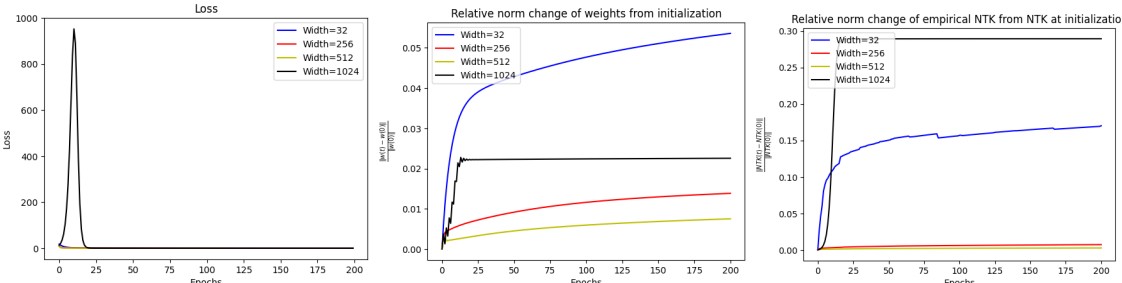

Figure 2: A GNN trained with the 0-1 Adjacency with self loops for a two layer GNN with a learning rate of 0.001 is not consistent with 1), 2) and 3).

|          | Wiki        | Cora       | Citeseer    |
|----------|-------------|------------|-------------|
| GCN-8    | 0.09, 0.025 | 0,14, 0.22 | 0.17, 0.16  |
| GCN-32   | 0.07, 0.32  | 0.14, 0.25 | 0.18, 0.2   |
| GCN-128  | 0.07, 0.45  | 0.15, 0.13 | 0.17, 0.16  |
| GCN-512  | 0.08, 0.49  | 0,16, 0.1  | 0.17, 0.16  |

Table 4: Overfitting with Gradient Descent, comma-seperated loss & accuracy

|          | Wiki       | Cora       | Citeseer    |
|----------|------------|------------|-------------|
| GCN-8    | 0.1, 0.15  | 0, 0.75    | 0.017, 0.57 |
| GCN-32   | 0.06, 0.08 | 0, 0.73    | 0, 0.6      |
| GCN-128  | 0.07, 0.45 | 0.15, 0.13 | 0.17, 0.16  |
| GCN-512  | 0.01, 0.73 | 0, 0.76    | 0, 0.625    |

Table 5: Overfitting with Adam Optimizer, comma-seperated loss & accuracy

## 8.3 GAT*GP and GAT*NTK with different nonlinearities

We compared different nonlinearities for the GAT*GP and GAT*NTK. GAT*GP/GAT*NTK-LINEAR have no nonlinearity. GAT*GP/GAT*NTK-2 have two nonlinearities $\sigma_2$ was the LeakyRelu as above and $\sigma_1$ is the Error-function[2]. We can see that adding two nonlinearities can improve the accuracy by 1%-2%.

|                 | Cora | Citeseer | Wiki |
|-----------------|------|----------|------|
| GAT*GP-LINEAR   | 0.79 | 0.72     | 0.74 |
| GAT*NTK-LINEAR  | 0.79 | 0.72     | 0.74 |
| GAT*GP          | 0.79 | 0.71     | 0.78 |
| GAT*NTK         | 0.79 | 0.71     | 0.77 |
| GAT*GP-2        | 0.79 | 0.71     | 0.78 |
| GAT*NTK-2       | 0.8  | 0.71     | 0.80 |

Table 6: Different GAT models with different nonlinearities.

---

[2] https://en.wikipedia.org/wiki/Error_function

## 8.4 CODE & ENVIRONMENT

All Experiments were conducted on an AMD Ryzen 7 3800X 8-Core Processor with hyperthreading with 32GB Ram. Experiments with † were conducted on an Intel(R) Xeon(R) Gold 6148 CPU @ 2.40GHz with 20 Cores without hyperthreading with 362GB of Ram. All Experiments were conducted using the CPU. The measurements for GAT*GP and GAT*NTK Pubmed, Squirrel and Crocodile are the datasets with 90% of edges removed via Effective Resistance. The GAT*GP and GAT*NTK code is highly optimized, making use of the Python Intel MKL Library which has an implementation of parallel Sparse Matrix multiplication whereas Pytorch and Scipy perform Sparse Matrix operations on one core only. In depth explanation and code is available at Github

|  | Cora/Citeseer/Wiki | Pubmed | Chameleon | Squirrel | Crocodile |
|---|---|---|---|---|---|
| Neural Nets | <1min | <2min | <2min | <3min | <3min |
| NNGP,NTK, GNNGP, GNTK SGNN, SGNTK | <2s | 2min | 1.15min | <3min | <3min |
| GAT*GP, GAT*NTK | 10min, 15min, 1h <16GB Ram | 25h, 60GB Ram† | 4h, 32GB Ram† | 1h, 32GB Ram† | 1h, 32GB Ram† |

## 8.5 EFFECTIVE RESISTANCE SPECTRAL GRAPH SPARSIFICATION

Recall that the GAT*GP and GAT*NTK require calculating the expression:

$$\psi(\Lambda^l) := \mathbb{E}\big[\sigma_1(v)\sigma_1(v^T)\big] \text{ with } v \sim \text{GP}\big(0, \sigma_c^2\gamma_A(\Lambda^{l-1})\big) \text{ and } \gamma_A(\Omega) := J_A\big(\begin{smallmatrix}\Omega & \Omega \\ \Omega & \Omega\end{smallmatrix}\big)J_A^T$$

and $J_A = \text{diag}(\text{vec}(A)) \text{concat}\,((\mathbf{1}_n \otimes I_n), (I_n \otimes \mathbf{1}_n))$ which comes from 3.5.

The expression $\gamma_A(\Omega) := J_A\big(\begin{smallmatrix}\Lambda^l & \Lambda^l \\ \Lambda^l & \Lambda^l\end{smallmatrix}\big)J_A^T$ will be $\in \mathbb{R}^{n^2 \times n^2}$ where $A \in \mathbb{R}^{n \times n}$. For Pubmed this will be a sparse $400,000 \times 400,000$ matrix. Calculating $\psi(\Lambda^l) := \mathbb{E}\big[\sigma_1(v)\sigma_1(v^T)\big]$ will again need multiple operations on the same matrix. Note that in the case of $\sigma_1$ being the Identitiy Function $\psi(\Lambda^l) = \gamma_A(\Omega)$. To still incorporate $\sigma_1$ in an efficient manner recall the definition of a two layer GAT Model:

$$G^1 = W^1 X$$
$$F^2 = \sigma_2(G^1\sigma_1(A \odot C)) \text{ with } C_{ij} = c_1^T G_i^1 + c_2^T G_j^1$$

with A being the 0-1 adjacency and with added self loops and $\sigma_1(0) = 0$ (.e.g. Relu, Sigmoid), results in $F^2 = \sigma_2(G^1(A \odot \sigma_1(C)))$. The corresponding Kernel would be

$$\psi(\Lambda^l) := \gamma_A\bigg(\mathbb{E}\big[\sigma_1(v)\sigma_1(v^T)\big]\bigg) \text{ with } v \sim \text{GP}\big(0, \sigma_c^2\Lambda^{l-1}\big)$$

This way we $\mathbb{E}\big[\sigma_1(v)\sigma_1(v^T)\big]$ will now be much more efficient as the resulting matrix will be $\in \mathbb{R}^{n \times n}$ instead $\in \mathbb{R}^{n^2 \times n^2}$. Calculating $\gamma_A(\Omega)$ for large graphs/matrices is still difficult and that is why we used Spectral Graph Sparsification method, namely Effective Resistance on PubMed, Squirrel and Crocodile for GAT*GP and GAT*NTK. EffectiveResistence (Spielman and Srivastava, 2011) was successfully applied for GNNs and GATs and one could show that it is possible to remove as much as 90% of the edges of a graph without sacrifice much performance and improving memory and runtime (Srinivasa et al., 2020). Effective Resistance takes an unweighted 0-1 adjacency matrix and returns a sparsified (i.e. graph with fewer edges) weighted adjacency matrix such that the Laplacians of these adjacency matrices are close in spectral norm. In our use case, we apply Effective Resistance to the 0-1 adjacency matrix and then replace each edge weight with the value one, so it is again a 0-1 adjacency matrix. This way we could improve speed especially for

Pubmed considering it took around 30 hours and 60GB Ram despite the fact that 90% of the edges had been removed. The following tables demonstrate Effective Resistance on GNNGP and GNTK Models. We run experiments with 50% and 90 % of the edges removed. There is a performance penalty for removing edges, but the difference in Cora, Citeseer, and Pubmed between removing 90% or 50% is small.

|  | Cora | Citeseer | Pubmed | Wiki | Chameleon | Squirrel | Crocodile |
|---|---|---|---|---|---|---|---|
| GNNGP | 0.83 | 0.71 | 0.80 | 0.78 | 0.64 | 0.48 | 0.78 |
| GNTK | 0.83 | 0.72 | 0.79 | 0.79 | 0.68 | 0.51 | 0.79 |
| GNNGP 0.5 | 0.77 | 0.66 | 0.75 | 0.70 | 0.52 | 0.4 | 0.45 |
| GNTK 0.5 | 0.76 | 0.67 | 0.75 | 0.71 | 0.52 | 0.46 | 0.15 |
| GNNGP 0.9 | 0.77 | 0.69 | 0.77 | 0.40 | 0.45 | 0.28 | 0.42 |
| GNTK 0.9 | 0.78 | 0.69 | 0.77 | 0.40 | 0.45 | 0.27 | 0.38 |

Table 7: Performane with Graph Spectral Sparsification known as Effective Resistance

## 8.6 Role of bias $\sigma_b^2$ for Regression

For Regression experiments with GPs and Kernels we set $\sigma_b^2 = 0.1$ and for the GNNs for Regression we added batch normalization, so we could reproduce the results from Niu et al. (2023). We are going to explore the performance for 1) Having no bias parameter for the GNNs 2) Setting $\sigma_b^2 = 0$ for the GP and NTK with the Sigmoid nonlinearity 3) Setting $\sigma_b^2 = 0$ for the GP and NTK with the Relu nonlinearity 4) $\sigma_b^2 = 0$ for GNNGP and GNTK with the Relu nonlinearity. For Classification, trying different $\sigma_b^2$ or having the models without bias term did not have the same effect as for Regression (i.e. no large decrease in performance). For Regression, the bias term for the Neural Nets as well as the $\sigma_b^2$ seems to play a crucial role and by leaving it out the performance worsens in practically every example (including negative $R^2$-Scores).

|  | Chameleon | Squirrel | Crocodile |
|---|---|---|---|
| FCN (No Bias) | 0.54 | 0.38 | 0.75 |
| GNN (No Bias) | -0.13 | -1.51 | 0.17 |
| NNGP (Sigmoid) ($\sigma_b = 0$) | 0.46 | 0.40 | 0.75 |
| NTK (Sigmoid) ($\sigma_b = 0$) | 0.46 | 0.40 | 0.75 |
| NNGP (Relu)($\sigma_b = 0$) | -1.59 | -7.917 | 0.726 |
| NTK (Relu) ($\sigma_b = 0$) | -1.481 | -7.314 | 0.725 |
| GNNGP ($\sigma_b = 0$) | 0.175 | -0.817 | 0.299 |
| GNN ($\sigma_b = 0$) | 0.202 | -0.757 | 0.34 |

Table 8: Role of bias $\sigma_b^2$ for Regression

## 8.7 Recap: Fully-connected Deep Nets

This section summarizes results on infinite width Fully Connected Networks as our derivations for infinite width Graph Neural Networks will be reduced to the fully connected derivation. We start with the definition and the derivatives.

**Definition 8.1** (Fully-connected Deep Neural Nets).

$$F^h(X) = \frac{\sigma_w}{\sqrt{d_{h-1}}} W^h G^{h-1}(X) + \sigma_b B^h \in \mathbb{R}^{d_h \times n} \tag{42}$$

$$with \ B^h = b^h \otimes \mathbf{1}_n^T \ and \ b^h \in \mathbb{R}^{d_h} \tag{43}$$

$$G^h(X) = \sigma(F^h(X)) \tag{44}$$
$$G^0 = X \in \mathbb{R}^{d_0 \times n} \tag{45}$$

*The Weights are initialized as $W^h \sim N(0,1)$ and $b^h \sim N(0,1)$.*
*The final Network is $F^L(X) = \frac{\sigma_w}{\sqrt{d_{L-1}}} W^L G^{L-1}(X) + \sigma_b B^L \in \mathbb{R}^{d_L \times n}$*

### 8.7.1 DERIVATIVES

The derivatives will be necessary for the closed form of the NTK.
The derivative of $F(\theta, X)$ with respect $W^l$ and $b^l$ can be derived using the chain rule.

$$\frac{\partial \operatorname{vec}(F(\theta, X))}{\partial \operatorname{vec}(W^l)} = \frac{\partial \operatorname{vec}(F^L)}{\partial \operatorname{vec}(G^{L-1})} \frac{\partial \operatorname{vec}(G^{L-1})}{\partial \operatorname{vec}(F^{L-1})} \cdots \frac{\partial \operatorname{vec}(F^{l+1})}{\partial \operatorname{vec}(G^l)} \frac{\partial \operatorname{vec}(G^l)}{\partial \operatorname{vec}(F^l)} \frac{\partial \operatorname{vec}(F^l)}{\partial \operatorname{vec}(W^l)}$$

Using

$$\frac{\partial \operatorname{vec}(F^h)}{\partial \operatorname{vec}(G^{h-1})} = \frac{\sigma_w}{\sqrt{d_{h-1}}} (I_n \otimes W^h)$$

$$\Sigma^h := \frac{\partial \operatorname{vec}(G^h)}{\partial \operatorname{vec}(F^h)} = \operatorname{diag}(\operatorname{vec}(\dot\sigma(F^h)))$$

$$\frac{\partial \operatorname{vec}(F^h)}{\partial \operatorname{vec}(W^h)} = \frac{\sigma_w}{\sqrt{d_{h-1}}} (G^{h-1^T} \otimes I_{d_h})$$

$$\frac{\partial \operatorname{vec}(F^h)}{\partial b^h} = \sigma_b (\mathbf{1}_n \otimes I_{d_h})$$

Combining the derivatives with the chain rule the final expression is:

$$\frac{\partial \operatorname{vec}(F(X))}{\partial \operatorname{vec}(W^l)} = \frac{\sigma_w}{\sqrt{d_{h-1}}} (I_n \otimes W^L) \Sigma^{L-1} \cdots \frac{\sigma_w}{\sqrt{d_{h-1}}} (I_n \otimes W^l) \Sigma^l \frac{\sigma_w}{\sqrt{d_{h-1}}} (G^{l-1^T} \otimes I_{d_h})$$

and similarly

$$\frac{\partial \operatorname{vec}(F(X))}{\partial b^l} = \frac{\sigma_w}{\sqrt{d_{h-1}}} (I_n \otimes W^L) \Sigma^{L-1} \cdots \frac{\sigma_w}{\sqrt{d_{h-1}}} (I_n \otimes W^l) \Sigma^l \sigma_b (\mathbf{1}_n \otimes I_{d_h})$$

Now we are going to show how the output of an FCN during training with $l_2$ loss with Gradient Flow (i.e. infinitesimally step size for Gradient Descent) can be described by the following differential equation:

$$\frac{du(\theta(t))}{dt} = \frac{\partial u(\theta(t))}{\partial \theta} \left( \frac{\partial u(\theta(t))}{\partial \theta} \right)^T (u(\theta(t)) - \operatorname{vec}(Y)) \tag{46}$$

with $u(\theta(t)) := \operatorname{vec}(F(\theta(t), X))$ and $\theta$ being all trainable parameters flattened to a vector. This equation nothing else then Kernel Regression under Gradient Flow. When all widths of the hidden Layers of $F(\theta(t), X))$ go to infinity, $\frac{\partial u(\theta(t))}{\partial \theta} \left( \frac{\partial u(\theta(t))}{\partial \theta} \right)^T = \Theta$ has a closed form expression which stays constant during training and the equation reduces to Kernel Regression. $\Theta$ is referred to as the NTK. In section 8.2 simulations confirm the NTK behavior for wide widths GNNs during training. For increased width during training, we can observe 1) the training loss approaches zero 2) the weights of the GNN stay close to its initialization (measured in relative Frobenius norm) and the the empirical NTK, (i.e. $\frac{\partial u(\theta(t))}{\partial \theta} \left( \frac{\partial u(\theta(t))}{\partial \theta} \right)^T$) stays close to its initialization during training. The proof of 1) consists of showing that the empirical NTK

stays positive definite during training (Du et al., 2019c). Proof of 2) consists of first showing that during initialization the empirical NTK is close to the closed form formula for the widths going to infinity. After that one has to show that during training the weights do not change by a lot using Gradient Flow Dynamics and finally proving: 2) → 3) (Jacot et al., 2018; Arora et al., 2019a). We are going to formally define FCNs and rederive the NNGP and NTK closed form expressions. The weight matrices are $W^h \in \mathbb{R}^{d_h \times d_{h-1}}$ and data matrix is $X \in \mathbb{R}^{d_0 \times n}$ where $d_0$ is the feature dimension of the data and $n$ is the number of data samples. $B$ denotes the rank-one bias matrix. $\sigma$ is a non-linearity (e.g. Relu) and is applied elementwise. The following lemma describes how the output of a Fully Connected Neural Net trained with Gradient Descent with an infinitesimally small learning rate can be interpreted as Kernel Regression with the Neural Tangent Kernel.

**Lemma 8.1** (From (Arora et al., 2019a)). *Consider minimizing the squared loss $\ell(\theta)$ by Gradient Descent with infinitesimally small learning rate[3]: $\frac{d\theta(t)}{dt} = -\nabla \ell(\theta(t))$. Then the output of the Network $F(\theta, X)$ evolves as:*

$$\frac{du(t)}{dt} = -H(t) \cdot \text{vec}(u(t) - Y)$$

*with $u(t) = F(\theta(t), X)$*

*and $H(t) = \frac{\partial \text{vec}(F(\theta(t), X)}{\partial \theta} \left( \frac{\partial \text{vec}(F(\theta(t), X)}{\partial \theta} \right)^T$*

*Proof.* The parameters $\theta$ evolve according to the differential equation

$$\frac{d\theta(t)}{dt} = -\nabla \ell(\theta(t)) = \left( \frac{\partial \text{vec}(F(\theta(t), X)}{\partial \theta} \right)^T \text{vec}(F(\theta(t), X) - Y)^T$$

where $t > 0$ is a continous time index. Using this equation, the evolution of the Network output $F(\theta(l), X)$ can be written as

$$\frac{dF(\theta(t), X}{dt} = \frac{\partial \text{vec}(F(\theta(t), X)}{\partial \theta} \left( \frac{\partial \text{vec}(F(\theta(t), X)}{\partial \theta} \right)^T \text{vec}(F(\theta(t), X) - Y)^T$$

Rewriting it using $\mathbf{u}(t) = \text{vec}(F(\theta(t), X))$

$$\frac{d\mathbf{u}(t)}{dt} = -\mathbf{H}(t) \cdot \text{vec}(\mathbf{u}(t) - Y)$$

$\square$

When the width of the network is allowed to go to infinity, it can be shown that the matrix $H(t)$ remains **constant** during training, i.e. equal to $H(0)$. Moreover, under the random initialization of the parameters defined in 8.1, the matrix $H(0)$ converges in probability to a certain deterministic Kernel matrix $H^*$ which is called the **Neural Tangent Kernel** $\Theta$ evaluated on the training data. If $H(t) = H^*$ for all $t$, then equation becomes

$$\frac{du(t)}{dt} = -H^* \cdot \text{vec}(u(t) - Y)$$

which is identical to the dynamics of **Kernel Regression** under Gradient Glow, for which at time $t \to \infty$ the final prediction function is

$$F^*(X_{test}) = H^*_{test,train}(H^*_{train,train})^{-1} y$$

---

[3] https://en.wikipedia.org/wiki/Euler_method

In the transductive setting $H^*_{train,train}$ implicitly accounts for incorporating the test data during training. The calculate the closed form expression for the NTK of a FCN of depth $l$ we have to simplify:

$$\Theta^l = \frac{\partial \operatorname{vec}(F^l(\theta, X))}{\partial \theta} \left( \frac{\partial \operatorname{vec}(F^l(\theta, X))}{\partial \theta} \right)^T$$

Using our model definition, this results in calculating

$$\Theta^L = \sum_{h=1}^{H} \frac{\partial \operatorname{vec}(F^L(\theta, X))}{\partial \operatorname{vec}(W^h)} \left( \frac{\partial \operatorname{vec}(F(\theta, X))}{\partial \operatorname{vec}(W^h)} \right)^T + \frac{\partial \operatorname{vec}(F(\theta, X))}{\partial \operatorname{vec}(b^h)} \left( \frac{\partial \operatorname{vec}(F^L(\theta, X))}{\partial \operatorname{vec}(b^h)} \right)^T \tag{47}$$

### 8.7.2 Gaussian Process

**Theorem 8.1** (Neural Network GP).
*If all the Weight dimensions $d_{L-1}, d_{L-1}, ..., d_1$ other than the input and the output dimensions of a Neural Network $F(X)^L$ successively go to infinity then $\operatorname{vec}(F(X)^L) \sim GP(0, \Lambda^{L-1} \otimes I_{d_L})$, with*

$$\Lambda^L = \sigma_w^2 \begin{pmatrix} \mathbb{E}[\sigma(u_1)\sigma(u_1)] & \mathbb{E}[\sigma(u_1)\sigma(u_2)] \dots & \mathbb{E}[\sigma(u_1)\sigma(u_n)] \\ \vdots & \ddots & \vdots \\ \mathbb{E}[\sigma(u_n)\sigma(u_1)] & \dots & \mathbb{E}[\sigma(u_n)\sigma(u_n)] \end{pmatrix} + \sigma_b^2 \tag{48}$$

$$\text{with } u \sim GP(0, \Lambda^{L-1}) \tag{49}$$

*$\Lambda^L$ can be written as the outer product of the random vector $u$*

$$\Lambda^L = \sigma_w^2 \mathbb{E}\left[\sigma(u)\sigma(u^T)\right] + \sigma_b^2 \tag{50}$$

*and the base case concludes the theorem*

$$\Lambda^0 = \frac{\sigma_w^2}{d_0} X^T X + \sigma_b^2 \tag{51}$$

The Integral $\mathbb{E}\left[\sigma(u)\sigma(u^T)\right]$ with $u$ being a multivariate Gaussian vector can be calculated in closed form for certain activation functions like Relu. Han et al. (2022) present an overview with closed form expressions for activation functions like Relu, Error function, Leaky Relu, Exponential, RBF, and others. For a list of efficient implementation see Novak et al. (2020). The proof is done by induction on the depth $l$ for $F^l(X)$. The Base Case is established on the fact that the output of $\operatorname{vec}(Y^1) = \operatorname{vec}(F(X)^1)$ is a weighted sum of zero mean multivariate Gaussian. The induction step is based on the fact that the output of $\operatorname{vec}(F^{l+1}(X))$ conditioned on $\operatorname{vec}(G^l(X))$ is again a Zero Mean Multivariate Gaussian. When the output of the previous layer goes to infinity, so $d_l \to \infty$, we will have an infinite sum which will converge to its mean by the Law of Large Numbers.

*Proof.* The matrix $\operatorname{Cov}(Y)_{IJ} \in \mathbb{R}^{d_1 \times d_1}$ will correspond to the matrix $\operatorname{Cov}(Y_{\cdot i}, Y_{\cdot j})$. And similarly $\mathbb{E}(Y)_I \in \mathbb{R}^{d_1}$ will be $\mathbb{E}(Y_{\cdot i})$. We will first derive the $\mathbb{E}(x)$ and $\operatorname{Cov}(x, x')$ with respect to a single, respectively two data samples and then derive the Expectation and Covariance for the whole dataset. Base Case for single datasample, respectively two data samples:

$$\mathbb{E}(Y)_I = \frac{\sigma_w}{\sqrt{d_0}} W^1 X_{\cdot i} + \sigma_b b^1 = 0$$

$$\operatorname{Cov}(Y)_{IJ} = \mathbb{E}\left[ \left( \frac{\sigma_w}{\sqrt{d_0}} W^1 X_{\cdot i} + \sigma_b b^1 \right) \left( \frac{\sigma_w}{\sqrt{d_0}} W^1 X_{\cdot j} + \sigma_b b^1 \right)^T \right]$$

$$= \frac{\sigma_w^2}{d_0} \mathbb{E}\left[W^1 X_{\cdot i} X_{\cdot j}^T W^{1^T}\right] + \sigma_b^2 \mathbb{E}\left[b^1 b^{1^T}\right]$$

$$= \frac{\sigma_w^2}{d_0} I_{d_1} \langle X_{\cdot i}, X_{\cdot j}\rangle + I_{d_1} \sigma_b^2$$

$$= I_{d_1} \left(\frac{\sigma_w^2}{d_0} \langle X_{\cdot i}, X_{\cdot j}\rangle + \sigma_b^2\right)$$

Base case for the complete dataset:

$$\mathbb{E}(\text{vec}(Y^1)) = \frac{\sigma_w}{\sqrt{d_0}} \text{vec}(W^1 X) + \sigma_b \text{vec}(B^1) = 0$$

$$\text{Cov}(\text{vec}(Y^1)) = \mathbb{E}\left[\left(\frac{\sigma_w}{\sqrt{d_0}} \text{vec}(W^1 X) + \sigma_b \text{vec}(B^1)\right)\left(\frac{\sigma_w}{\sqrt{d_0}} \text{vec}(W^1 X) + \sigma_b \text{vec}(B^1))^T\right)\right]$$

$$= \frac{\sigma_w^2}{d_0} \mathbb{E}\left[\text{vec}(W^1 X) \text{vec}(W^1 X)^T\right] + \sigma_b^2 \mathbb{E}\left[\text{vec}(B^1) \text{vec}(B^1)^T\right]$$

$$= \frac{\sigma_w^2}{d_0} \mathbb{E}\left[(I_n \otimes W^1) \text{vec}(X) \text{vec}(X)^T (I_n \otimes W^{1^T})\right] + \sigma_b^2 (\mathbb{1}_n \otimes I_{d_1})$$

$$= \left(\frac{\sigma_w^2}{d_0} X^T X + \sigma_b^2\right) \otimes I_{d_1}$$

The induction step for one, respectively two data samples.

$$\mathbb{E}(\mathbf{Y}^{l+1})_I = 0$$

$$\text{Cov}(\mathbf{Y}^{l+1})_{IJ} = \mathbb{E}\left[\left(\frac{\sigma_w}{\sqrt{d_l}} W^{l+1} G_{\cdot i}^l + \sigma_b b^{l+1}\right)\left(\frac{\sigma_w}{\sqrt{d_l}} W^{l+1} \sigma(Y_{\cdot i}^l) + \sigma_b b^{l+1}\right)^T\right]$$

$$= \frac{\sigma_w^2}{d_l} \mathbb{E}\left[W^{l+1} \sigma(Y_{\cdot i}^l) \sigma(Y_{\cdot j}^l)^T W^{l+1^T}\right] + \sigma_b^2 I_{d_l}$$

$$\frac{\sigma_w^2}{d_l} \mathbb{E}(W^{l+1} \sigma(Y_{\cdot i}^l) \sigma(Y_{\cdot j}^l W^{l+1^T})^T)_{uv} = \frac{\sigma_w^2}{d_l} \text{tr}(\sigma(Y_{\cdot i}^l) \sigma(Y_{\cdot j}^l)^T) \delta_{uv} \tag{51.1}$$

$$\delta_{uv} \frac{\sigma_w^2}{d_l} \sum_{n=1}^{d_l} \sigma(Y_{nj}^l) \sigma(Y_{ni}^l) \overset{d_l \to \infty}{\overset{P}{\longrightarrow}} \sigma_w^2 \delta_{uv} \underset{s,t \sim N(0, \begin{pmatrix} \Lambda_{ii}^{l-1} & \Lambda_{ij}^{l-1} \\ \Lambda_{ji}^{l-1} & \Lambda_{jj}^{l-1} \end{pmatrix})}{\mathbb{E}} \left[\sigma(s)\sigma(t)\right] \tag{51.2}$$

In (51.1) we have used the quadratic form for random variables. In (51.2) we have used the fact that $Y_{ni}^l$ and $Y_{mi}^l$ with $n \neq m$ are i.i.d. random variables. That also becomes apparent by observing that $\text{Cov}(Y_{\cdot i}^l, Y_{\cdot j}^l)$ is diagonal. And finally for the whole dataset:

$$\mathbb{E}(\text{vec}(Y^{l+1})) = \frac{\sigma_w}{\sqrt{d_l}} \text{vec}(W^{l+1} G^l) + \sigma_b \text{vec}(B^{l+1}) = 0$$

$$\text{Cov}(\text{vec}(Y^{l+1})) = \mathbb{E}\left[\left(\frac{\sigma_w}{\sqrt{d_l}} \text{vec}(W^{l+1} G^l) + \sigma_b \text{vec}(B^{l+1})\right)\left(\frac{\sigma_w}{\sqrt{d_l}} \text{vec}(W^{l+1} G^l) + \sigma_b \text{vec}(B^{l+1})^T\right)\right]$$

$$= \frac{\sigma_w^2}{d_l} \mathbb{E}\left[\text{vec}(W^{l+1} G^l) \text{vec}(G^{l^T} W^{l+1^T}\right] + \sigma_b^2 (\mathbb{1}_n \otimes I_{d_l})$$

$$= \frac{\sigma_w^2}{d_l} \mathbb{E}\left[(I_n \otimes W^{l+1}) \text{vec}(G^l) \text{vec}(G^l)^T (I_n \otimes W^{l+1^T})\right] + \sigma_b^2 (\mathbb{1}_n \otimes I_{d_{l+1}}) \tag{51.2}$$

$$\frac{\sigma_w^2}{d_l}\mathbb{E}\big[(I_n \otimes W^{l+1})\operatorname{vec}(G^l)\operatorname{vec}(G^l)^T(I_n \otimes W^{l+1^T})\big] + \sigma_b^2(\mathbb{1}_n \otimes I_{d_l})$$

$$= \left(\sigma_w^2 \begin{pmatrix} \mathbb{E}\big[\sigma(u_1)\sigma(u_1)\big] & \mathbb{E}\big[\sigma(u_1)\sigma(u_2)\big] \dots & \mathbb{E}\big[\sigma(u_1)\sigma(u_n)\big] \\ \vdots & \ddots & \vdots \\ \mathbb{E}\big[\sigma(u_n)\sigma(u_1)\big] & \dots & \mathbb{E}\big[\sigma(u_n)\sigma(u_n)\big] \end{pmatrix} + \sigma_b^2\right) \otimes I_{d_{l+1}}$$

with $u \sim N(0, \Lambda^{L-1})$

In (51.2) we have used the property of the Kronecker Product, namely $\operatorname{vec}(AXB) = (B^T \otimes A)\operatorname{vec}(X)$.  □

### 8.7.3 NEURAL TANGENT KERNEL

**Theorem 8.2** (NTK).
*If all the Weight dimensions $d_{L-1}, d_{L-1}, ..., d_1$ other than the input and the output dimensions of a Neural Network $F(X)^L$ successively go to infinity, the NTK has a closed form, namely $\Theta^L \otimes I_{d_l}$ with*

$$\Theta^L = \Lambda^{L-1} + \left(\dot{\Lambda}^{L-1} \odot \Theta^{L-1}\right) \tag{52}$$

$$\Lambda^L = \sigma_w^2\mathbb{E}\left[\sigma(u)\sigma(u^T)\right] + \sigma_b^2 \tag{53}$$

$$\dot{\Lambda}^L = \sigma_w^2\mathbb{E}\left[\dot{\sigma}(u)\dot{\sigma}(u^T)\right] + \sigma_b^2 \tag{54}$$

*with $u \sim N(0, \Lambda^{L-1})$*

$$\Theta^1 = \Lambda^0 = \frac{\sigma_w^2}{d_0}X^TX + \sigma_b^2 \tag{55}$$

*Proof.* Proof is by induction, similar to the NNGP derivation.
Base Case:

$$\Theta^1 = \Lambda^1 = \frac{\partial \operatorname{vec}(F^1(\theta, X))}{\partial \operatorname{vec}(W^1)}\left(\frac{\partial \operatorname{vec}(F^1(\theta, X))}{\partial \operatorname{vec}(W^1)}\right)^T + \frac{\partial \operatorname{vec}(F^1(\theta, X))}{\partial \operatorname{vec}(b^1)}\left(\frac{\partial \operatorname{vec}(F^1(\theta, X))}{\partial \operatorname{vec}(b^1)}\right)^T$$

$$= (\frac{\sigma_w^2}{d_0}X^TX + \sigma_b^2) \otimes I_{d_1}$$

Induction Step:

$$\Theta^{l+1} = \sum_{h=1}^{l+1}\frac{\partial \operatorname{vec}(F^{l+1}(\theta, X))}{\partial \operatorname{vec}(W^h)}\left(\frac{\partial \operatorname{vec}(F^{l+1}(\theta, X))}{\partial \operatorname{vec}(W^h)}\right)^T$$

$$= \underbrace{\frac{\partial \operatorname{vec}(F^{l+1}(\theta, X))}{\partial \operatorname{vec}(W^{l+1})}\left(\frac{\partial \operatorname{vec}(F^{l+1}(\theta, X))}{\partial \operatorname{vec}(W^{l+1})}\right)^T + \frac{\partial \operatorname{vec}(F^{l+1}(\theta, X))}{\partial \operatorname{vec}(b^{l+1})}\left(\frac{\partial \operatorname{vec}(F^{l+1}(\theta, X))}{\partial \operatorname{vec}(b^{l+1})}\right)^T}_{B}$$

$$+ \underbrace{\sum_{h=1}^{l}\frac{\partial \operatorname{vec}(F^{l+1}(\theta, X))}{\partial \operatorname{vec}(W^h)}\left(\frac{\partial \operatorname{vec}(F^{l+1}(\theta, X))}{\partial \operatorname{vec}(W^h)}\right)^T + \frac{\partial \operatorname{vec}(F^{l+1}(\theta, X))}{\partial \operatorname{vec}(b^h)}\left(\frac{\partial \operatorname{vec}(F^{l+1}(\theta, X))}{\partial \operatorname{vec}(b^h)}\right)^T}_{\Gamma}$$

$$B = \frac{\sigma_w^2}{d_l}(G^{l^T}G \otimes I_{d_{l+1}}) + \sigma_b^2(\mathbb{1}_n \otimes I_{d_{l+1}}) = (\frac{\sigma_w^2}{d_l}G^{l^T}G^l + \sigma_b^2) \otimes I_{d_{l+1}}$$

$$(\frac{\sigma_w^2}{d_l}G^{l^T}G^l)_{ij} = \frac{\sigma_w^2}{d_l}\sum_m^{d_l}G_{im}^l G_{jm}^l \overset{d_l \to \infty}{\underset{P}{\longrightarrow}} \sigma_w^2 \underset{u,v \sim N(0, \begin{pmatrix} \Lambda_{ii}^{l-1} & \Lambda_{ij}^{l-1} \\ \Lambda_{ji}^{l-1} & \Lambda_{jj}^{l-1} \end{pmatrix})}{\mathbb{E}} [\sigma(u)\sigma(v)]$$

Where we used the fact that the output of the previous layer goes to infinity (so $d_l \to \infty$)

$$\implies B = \Lambda^L \otimes I_{d_{l+1}}$$

$$\Gamma = \sum_{h=1}^{l} \left(\frac{\partial \operatorname{vec}(F^{l+1}(\theta,X))}{\partial \operatorname{vec}(G^l)}\frac{\partial \operatorname{vec}(G^l)}{\partial \operatorname{vec}(F^l)}\frac{\partial \operatorname{vec}(F^l(\theta,X))}{\partial \operatorname{vec}(W^l)}\right)\left(\frac{\partial \operatorname{vec}(F^{l+1}(\theta,X))}{\partial \operatorname{vec}(G^l)}\frac{\partial \operatorname{vec}(G^l)}{\partial \operatorname{vec}(F^l)}\frac{\partial \operatorname{vec}(F^l(\theta,X))}{\partial \operatorname{vec}(W^l)}\right)^T$$

$$+ \left(\frac{\partial \operatorname{vec}(F^{l+1}(\theta,X))}{\partial \operatorname{vec}(G^l)}\frac{\partial \operatorname{vec}(G^l)}{\partial \operatorname{vec}(F^l)}\frac{\partial \operatorname{vec}(F^l(\theta,X))}{\partial \operatorname{vec}(b^l)}\right)\left(\frac{\partial \operatorname{vec}(F^{l+1}(\theta,X))}{\partial \operatorname{vec}(G^l)}\frac{\partial \operatorname{vec}(G^l)}{\partial \operatorname{vec}(F^l)}\frac{\partial \operatorname{vec}(F^l(\theta,X))}{\partial \operatorname{vec}(b^l)}\right)^T$$

$$= \frac{\partial \operatorname{vec}(F^{l+1}(\theta,X))}{\partial \operatorname{vec}(G^l)}\frac{\partial \operatorname{vec}(G^l)}{\partial \operatorname{vec}(F^l)}(\Theta^l \otimes I_{d_l})\left(\frac{\partial \operatorname{vec}(F^{l+1}(\theta,X))}{\partial \operatorname{vec}(G^l)}\frac{\partial \operatorname{vec}(G^l)}{\partial \operatorname{vec}(F^l)}\right)^T$$

$$= \frac{\sigma_w^2}{d_l}(I_n \otimes W^{l+1})\Sigma^l(\Theta^l \otimes I_d)\Sigma^l(I_n \otimes W^{l+1^T})$$

$$T_{IJ} = \left(\frac{\sigma_w^2}{d_l}(I_n \otimes W^{l+1})\Sigma^l(\Theta^l \otimes I_d)\Sigma^l(I_n \otimes W^{l+1^T})\right)_{IJ} = \frac{\sigma_w^2}{d_l}W^{l+1}\Sigma_{II}^l\Theta_{ij}^l\Sigma_{JJ}^l W^{l+1^T}$$

$$= \Theta_{ij}^l\frac{\sigma_w^2}{d_l}W^{l+1}\Sigma_{II}^l\Sigma_{JJ}^l W^{l+1^T}$$

$$(\frac{\sigma_w^2}{d_l}W^{l+1}\Sigma_{II}^l\Sigma_{JJ}^l W^{l+1^T})_{uv} = \frac{\sigma_w^2}{d_l}\sum_{st}^{d_l}\Sigma_{II_{ss}}^l\Sigma_{JJ_{tt}}^l W_{us}W_{vt} \overset{d_l \to \infty}{\underset{P}{\longrightarrow}} \delta_{uv}\sigma_w^2 \underset{s,t \sim N(0, \begin{pmatrix} \Lambda_{ii}^{l-1} & \Lambda_{ij}^{l-1} \\ \Lambda_{ji}^{l-1} & \Lambda_{jj}^{l-1} \end{pmatrix})}{\mathbb{E}} [\dot{\sigma}(s)\dot{\sigma}(t)]$$

Convergence in probability follows from the Law of Large Numbers. The righthandside follows from the definition of $\Sigma^l$ and the fact that the output of layer $l-1$ in its infinite width is a Gaussian Process.

$$\implies T = (\Theta \odot \dot{\Lambda}^L) \otimes I_{d_{l+1}}$$

$$\implies \Theta^{l+1} = B + \Gamma = \Lambda^l + \left(\dot{\Lambda}^l \odot \Theta^l\right)$$

$\square$

The NTK Closed Form can be simplified by expanding the recursion and reordering the terms.

**Corollary 8.1** (Non-recursive NTK Formula (Equation 9 Arora et al., 2019a))**.**

$$\dot{\Lambda}^{L+1} := \mathbb{1}_n$$

$$\Theta^L = \sum_{h=2}^{L+1}\left(\Lambda^{h-1} \odot (\dot{\Lambda}^h \odot \dot{\Lambda}^{h+1} \odot \cdots \odot \dot{\Lambda}^{L+1})\right) \otimes I_{d_L}$$

## 8.8 NTK & GP FOR GRAPH NEURAL NETWORK

We copied the definitions of the Graph Neural Network from the main section.

**Definition 3.1** (Graph Neural Network).

$$F^h(X) = \left(\frac{\sigma_w}{\sqrt{d_{h-1}}}W^h G^{h-1}(X) + \sigma_b B^h\right)A^T \in \mathbb{R}^{d_h \times n} \text{ with } B^h = b^h \otimes \mathbf{1}_n^T \text{ and } b^h \in \mathbb{R}^{d_h} \quad (8)$$

$$G^h(X) = \sigma(F^h(X)) \text{ and } G^0 = X \in \mathbb{R}^{d_0 \times n} \quad (9)$$

*The weights are initialized as* $W^h \sim N(0,1)$ *and* $b^h \sim N(0,1)$.

*The final network of depth L is* $F^L(X) = \left(\frac{\sigma_w}{\sqrt{d_{L-1}}}W^L G^{L-1}(X) + \sigma_b B^L\right)A^T \in \mathbb{R}^{d_L \times n}$

### DERIVATIVES

The derivative of $F(\theta, X)$ with respect to $W^l$ is

$$\frac{\partial \operatorname{vec}(F(\theta,X))}{\partial \operatorname{vec}(W^l)} = \frac{\partial \operatorname{vec}(F^L)}{\partial \operatorname{vec}(G^{L-1})}\frac{\partial \operatorname{vec}(G^{L-1})}{\partial \operatorname{vec}(F^{L-1})}\cdots\frac{\partial \operatorname{vec}(F^{l+1})}{\partial \operatorname{vec}(G^l)}\frac{\partial \operatorname{vec}(G^l)}{\partial \operatorname{vec}(F^l)}\frac{\partial \operatorname{vec}(F^l)}{\partial \operatorname{vec}(W^l)}$$

Using

$$\frac{\partial \operatorname{vec}(F^h)}{\partial \operatorname{vec}(G^{h-1})} = \frac{\sigma_w}{\sqrt{d_{h-1}}}(A \otimes W^h)$$

$$\Sigma^h := \frac{\partial \operatorname{vec}(G^h)}{\partial \operatorname{vec}(F^h)} = \operatorname{diag}(\operatorname{vec}(\dot{\sigma}(F^h)))$$

$$\frac{\partial \operatorname{vec}(F^h)}{\partial \operatorname{vec}(W^h)} = \frac{\sigma_w}{\sqrt{d_{h-1}}}(AG^{h-1^T} \otimes I_{d_h})$$

$$\frac{\partial \operatorname{vec}(F^h)}{\partial b^h} = \sigma_b(A\mathbf{1}_n \otimes I_{d_h})$$

### 8.8.1 GAUSSIAN PROCESS (THEOREM 3.1)

**Theorem 3.1** (GNN Gaussian Process). *If all the weight dimensions of the hidden layers* $d_{L-1}, d_{L-1}, ..., d_1$ *of a GNN,* $F(X)^L$ *successively go to infinity, then* $\operatorname{vec}(F(X)^L) \sim GP(0, (A\Lambda^{L-1}A^T) \otimes I_{d_L})$ *with*

$$\Lambda^L = \sigma_w^2 \mathbb{E}\left[\sigma(u)\sigma(u^T)\right] + \sigma_b^2 \quad (10)$$

*with* $u \sim N(0, A\Lambda^{L-1}A^T)$

$$\Lambda^0 = \frac{\sigma_w^2}{d_0}X^T X + \sigma_b^2 \quad (11)$$

*Proof.* Proof is by induction. We define $\mathbf{Y}^L = F^L(X)$.

$$\mathbb{E}(\operatorname{vec}(Y^1)) = \frac{\sigma_w}{\sqrt{d_0}}\operatorname{vec}\left(\left(\frac{\sigma_w}{\sqrt{d_0}}W^1 X + \sigma_b B^1\right)A^T\right) = 0$$

$$\operatorname{Cov}(\operatorname{vec}(Y^1)) = \operatorname{Cov}\left[(A \otimes I_{d_1})\operatorname{vec}(\frac{\sigma_w}{\sqrt{d_0}}W^1 X + \sigma_b B^1)\right]$$

$$= (A \otimes I_{d_1})\mathbb{E}\big[\mathrm{vec}(\frac{\sigma_w}{\sqrt{d_0}}W^1 X A^T + \sigma_b B^1)\big](A^T \otimes I_{d_1})$$

$$= (A\Lambda^1 A^T) \otimes I_{d_1}$$

The Base Case is reduced to the Base Case of NNGP using a property of the Kronecker Product, namely $(A \otimes B)(C \otimes D) = (AC \otimes BD)$.

Induction Step is again just an application of the Kronecker Property and follows the NNGP proof in 8.1.

$$\mathbb{E}(\mathrm{vec}(Y^{l+1})) = 0$$
$$\mathrm{Cov}(\mathrm{vec}(Y^{l+1})) = (A\Lambda^{l+1}A^T) \otimes I_{d_{l+1}}$$

$\square$

### 8.8.2 GRAPH NEURAL TANGENT KERNEL (THEOREM 3.2)

**Theorem 3.2** (GNN NTK).
*If all the weight dimensions of the hidden layers $d_{L-1}, d_{L-1}, ..., d_1$ of a GNN, $F(X)^L$ successively go to infinity, the NTK has a closed form expression $\Theta^L \otimes I_{d_L}$, with*

$$\Theta^L = A\left(\Lambda^{L-1} + \left(\dot{\Lambda}^{L-1} \odot \Theta^{L-1}\right)\right)A^T \text{ with} \tag{12}$$

$$\Lambda^L = \sigma_w^2 \mathbb{E}\left[\sigma(u)\sigma(u^T)\right] + \sigma_b^2 \tag{13}$$

$$\dot{\Lambda}^L = \sigma_w^2 \mathbb{E}\left[\dot{\sigma}(u)\dot{\sigma}(u^T)\right] + \sigma_b^2 \tag{14}$$

*with $u \sim N(0, A\Lambda^{L-1}A^T)$*

$$\Theta^1 = \Lambda^0 = \sigma_w^2 X^T X + \sigma_b^2 \tag{15}$$

*Proof.* Proof is by induction, similar to the NNGP derivation.
Base Case:

$$\Theta^1 = \Lambda^1 = \frac{\partial \mathrm{vec}(F^1(\theta, X))}{\partial \mathrm{vec}(W^1)}\left(\frac{\partial \mathrm{vec}(F^1(\theta, X))}{\partial \mathrm{vec}(W^1)}\right)^T + \frac{\partial \mathrm{vec}(F^1(\theta, X))}{\partial \mathrm{vec}(b^1)}\left(\frac{\partial \mathrm{vec}(F^1(\theta, X))}{\partial \mathrm{vec}(b^1)}\right)^T$$

$$= \left(A\big(\frac{\sigma_w^2}{d_0}X^T X + \sigma_b^2\big)A^T\right) \otimes I_{d_1}$$

Induction Step:

$$\Theta^{l+1} = \sum_{h=1}^{l+1} \frac{\partial \mathrm{vec}(F^{l+1}(\theta, X))}{\partial \mathrm{vec}(W^h)}\left(\frac{\partial \mathrm{vec}(F^{l+1}(\theta, X))}{\partial \mathrm{vec}(W^h)}\right)^T$$

$$= \underbrace{\frac{\partial \mathrm{vec}(F^{l+1}(\theta, X))}{\partial \mathrm{vec}(W^{l+1})}\left(\frac{\partial \mathrm{vec}(F^{l+1}(\theta, X))}{\partial \mathrm{vec}(W^{l+1})}\right)^T + \frac{\partial \mathrm{vec}(F^{l+1}(\theta, X))}{\partial \mathrm{vec}(b^{l+1})}\left(\frac{\partial \mathrm{vec}(F^{l+1}(\theta, X))}{\partial \mathrm{vec}(b^{l+1})}\right)^T}_{B}$$

$$+ \underbrace{\sum_{h=1}^{l} \frac{\partial \mathrm{vec}(F^{l+1}(\theta, X))}{\partial \mathrm{vec}(W^h)}\left(\frac{\partial \mathrm{vec}(F^{l+1}(\theta, X))}{\partial \mathrm{vec}(W^h)}\right)^T + \frac{\partial \mathrm{vec}(F^{l+1}(\theta, X))}{\partial \mathrm{vec}(b^h)}\left(\frac{\partial \mathrm{vec}(F^{l+1}(\theta, X))}{\partial \mathrm{vec}(b^h)}\right)^T}_{\Gamma}$$

$$B = \left(A\left(\frac{\sigma_w^2}{d_l}G^{l^T}G^l + \sigma_b^2\right)A^T\right) \otimes I_{d_{l+1}}$$

We can now proceed similar to the previous NTK derivations (see 8.2) and therefore will skip the parts which stay the same.

$$\implies B = A\left(\Lambda^L \otimes I_{d_{l+1}}\right) A^T$$

$$\Gamma = \sum_{h=1}^{l} \left(\frac{\partial \operatorname{vec}(F^{l+1}(\theta, X))}{\partial \operatorname{vec}(G^l)} \frac{\partial \operatorname{vec}(G^l)}{\partial \operatorname{vec}(F^l)} \frac{\partial \operatorname{vec}(F^l(\theta, X))}{\partial \operatorname{vec}(W^l)}\right) \left(\frac{\partial \operatorname{vec}(F^{l+1}(\theta, X))}{\partial \operatorname{vec}(G^l)} \frac{\partial \operatorname{vec}(G^l)}{\partial \operatorname{vec}(F^l)} \frac{\partial \operatorname{vec}(F^l(\theta, X))}{\partial \operatorname{vec}(W^l)}\right)^T$$

$$+ \left(\frac{\partial \operatorname{vec}(F^{l+1}(\theta, X))}{\partial \operatorname{vec}(G^l)} \frac{\partial \operatorname{vec}(G^l)}{\partial \operatorname{vec}(F^l)} \frac{\partial \operatorname{vec}(F^l(\theta, X))}{\partial \operatorname{vec}(b^l)}\right) \left(\frac{\partial \operatorname{vec}(F^{l+1}(\theta, X))}{\partial \operatorname{vec}(G^l)} \frac{\partial \operatorname{vec}(G^l)}{\partial \operatorname{vec}(F^l)} \frac{\partial \operatorname{vec}(F^l(\theta, X))}{\partial \operatorname{vec}(b^l)}\right)^T$$

$$= \frac{\sigma_w^2}{d_l}(A \otimes W^{l+1})\Sigma^l(\Theta^l \otimes I_d)\Sigma^l(A^T \otimes W^{l+1^T})$$

$$= \frac{\sigma_w^2}{d_l}(A \otimes I_{d_l})(I_n \otimes W^{l+1})\Sigma^l(\Theta^l \otimes I_d)\Sigma^l(I_n \otimes W^{l+1^T}(A^T \otimes I_{d_l}))$$

We reduced the proof to the NTK proof using property of the Kronecker Product in the last line.

$$\implies T = (A(\Theta \odot \dot{\Lambda}^L)A^T) \otimes I_{d_{l+1}}$$

$$\square$$

### 8.9 NTK & GP FOR GRAPH NEURAL NETWORK WITH SKIP-CONCATENATE CONNECTIONS

The model definition is copied from Section 3.2 for the readers convenience.

**Definition 3.2** (Skip-Concatenate Graph Neural Network).

$$F^h(X) = \left(\frac{\sigma_w}{\sqrt{2d_{h-1}}} W^h G^{h-1}(X) + \sigma_b B^h\right) A^T \text{ for } h > 1 \tag{16}$$

$$F^1(X) = \left(\frac{\sigma_w}{\sqrt{d_1}} W^1 G^0(X) + \sigma_b B^1\right) A^T \text{ with } B^h = b^h \otimes \mathbf{1}_n^T \text{ and } b^h \in \mathbb{R}^{d_h} \tag{17}$$

$$G^h(X) = \begin{pmatrix} \sigma(F^h(X)) \\ F^h(X) \end{pmatrix} \text{ ,so } G^h(X) \in \mathbb{R}^{2d_l \times n} \text{ and } G^0 = X \in \mathbb{R}^{d_0 \times n} \tag{18}$$

*The weights are initialized as $W^h \sim N(0, 1)$ and $b^h \sim N(0, 1)$.*

*The final Network of depth $L$ is $F^L(X) = \left(\frac{\sigma_w}{\sqrt{d_{L-1}}} W^L G^{L-1}(X) + \sigma_b B^L\right) A^T \in \mathbb{R}^{d_L \times n}$.*

The derivatives are similar to the Graph Neural Network derivatives (see 8.8). Only $\Sigma^h$ is different.

$$\Sigma^h := \frac{\partial \operatorname{vec}(G^h)}{\partial \operatorname{vec}(F^h)} = \begin{pmatrix} \operatorname{diag}(\operatorname{vec}\dot{\sigma}(F^h_{\cdot 1})) & 0 & 0 & \dots & 0 \\ I_{d_h} & 0 & 0 & \dots & 0 \\ 0 & \operatorname{diag}(\operatorname{vec}\dot{\sigma}(F^h_{\cdot 2})) & 0 & \dots & 0 \\ 0 & I_{d_h} & 0 & \dots & \\ 0 & 0 & \ddots & & \\ \vdots & \vdots & \ddots & & \\ 0 & 0 & & \dots & \operatorname{diag}(\operatorname{vec}\dot{\sigma}(F^h_{\cdot n})) \\ 0 & 0 & 0 & \dots & I_{d_h} \end{pmatrix} \in \mathbb{R}^{2d_l n \times 2d_l n}$$

$\Sigma^h$ is block-diagonal, with each block $\Sigma_{II}^h = \begin{pmatrix} \text{diag}(\text{vec}\,\dot\sigma(F_{\cdot i}^h)) \\ I_{d_h} \end{pmatrix}$

### 8.9.1 S-GNN GAUSSIAN PROCESS (THEOREM 3.3)

**Theorem 3.3** (Skip-Concatenate GNN Gaussian Process).
*If all the weight dimensions of the hidden layers $d_{L-1}, d_{L-1}, ..., d_1$ of a Skip-GNN $F(X)^L$ successively go to infinity, then* $\text{vec}(F(X)^L) \sim GP(0, (A\Lambda^{L-1}A^T) \otimes I_{d_L})$, *with*

$$\Lambda^L = \sigma_w^2 \frac{1}{2}\left(\mathbb{E}\left[\sigma(u)\sigma(u^T)\right] + A\Lambda^{L-1}A^T\right) + \sigma_b^2 \tag{19}$$

*with* $u \sim N(0, A\Lambda^{L-1}A^T)$

$$\Lambda^0 = \frac{\sigma_w^2}{d_0}X^T X + \sigma_b^2 \tag{20}$$

*Proof.* Prove is by induction. We define $Y^L = F^L(X)$. The Base Case doesn't change, so

$$\mathbb{E}(\text{vec}(Y^1)) = 0$$
$$\text{Cov}(\text{vec}(Y^1)) = (A\Lambda^1 A^T) \otimes I_{d_1}$$

The Induction Step:

$$\mathbb{E}(\text{vec}(Y^{l+1})) = 0$$

$$\text{Cov}(\text{vec}(Y^{l+1})) = \text{Cov}\left[(A \otimes I_{d_{l+1}})\text{vec}(\frac{\sigma_w}{\sqrt{d_0}}W^{l+1}G^l A^T + \sigma_b B^{l+1})\right]$$

$$= (A \otimes I_{d_{l+1}})\text{Cov}\left[\text{vec}(\frac{\sigma_w}{\sqrt{d_0}}W^{l+1}G^l + \sigma_b B^1)\right](A^T \otimes I_{d_{l+1}})$$

$$= (A \otimes I_{d_{l+1}})\left(\frac{\sigma_w^2}{d_l}\mathbb{E}\left[(I_n \otimes W^{l+1})\text{vec}(G^l)\text{vec}(G^l)^T(I_n \otimes W^{l+1^T})\right] + \sigma_b^2(\mathbb{1}_n \otimes \mathbf{1}_{d_{l+1}})\right)(A^T \otimes I_{d_{l+1}})$$

$$\frac{\sigma_w^2}{2d_l}\mathbb{E}\left[(I_n \otimes W^{l+1})\text{vec}(G^l)\text{vec}(G^l)^T(I_n \otimes W^{l+1^T})\right]_{IJ_{uv}} = \frac{\sigma_w^2}{2d_l}\mathbb{E}\left[W_u(\text{vec}(G^l)\text{vec}(G^l)^T)_{IJ}W_v^T\right]$$

$Z := \left(\text{vec}(G^l)\text{vec}(G^l)^T\right)_{IJ}$ and $Z_{11}$ is the upper left block matrix of size $\mathbb{R}^{d_l \times d_l}$.

and $W_u^{l+1} = \left[\overset{1}{W}_u^{l+1}, \overset{2}{W}_u^{l+1}\right]$ with $\overset{1}{W}_u^{l+1}, \overset{2}{W}_u^{l+1} \in \mathbb{R}^{1 \times d_l}$, so we split $W_u^{l+1}$ into two parts, each of length $d_l$.

$$\frac{\sigma_w^2}{2d_l}\mathbb{E}\left[W_u(\text{vec}(G^l)\text{vec}(G^l)^T)_{IJ}W_v^T\right]$$

$$= \frac{\sigma_w^2}{2d_l}\mathbb{E}\left[\overset{1}{W}_u^{l+1}Z_{11}\overset{1}{W}_v^{l+1} + \overset{1}{W}_u^{l+1}Z_{12}\overset{2}{W}_v^{l+1} + \overset{2}{W}_u^{l+1}Z_{21}\overset{1}{W}_v^{l+1} + \overset{2}{W}_u^{l+1}Z_{22}\overset{2}{W}_v^{l+1}\right]$$

$$= \delta_{uv}\frac{\sigma_w^2}{2d_l}\mathbb{E}\left[\overset{1}{W}_u^{l+1}Z_{11}\overset{1}{W}_v^{l+1} + \overset{2}{W}_u^{l+1}Z_{22}\overset{2}{W}_v^{l+1}\right]$$

$$= \delta_{uv} \frac{\sigma_w^2}{2d_l} \left( \sum_s^{d_l} \sigma(Y^l)_{is} \sigma(Y^l)_{js} + \sum_s^{d_l} (Y_{is}^l Y_{js}^l) \right) = \delta_{uv} \frac{\sigma_w^2}{2d_l} \left( \sum_s^{d_l} \sigma(Y^l)_{is} \sigma(Y^l)_{js} + (Y_{is}^l Y_{js}^l) \right)$$

$$\xrightarrow{d_l \xrightarrow{P} \infty} \delta_{uv} \frac{1}{2} \left( \underset{s,t \sim N(0, \begin{pmatrix} \Lambda_{ii}^{l-1} & \Lambda_{ij}^{l-1} \\ \Lambda_{ji}^{l-1} & \Lambda_{jj}^{l-1} \end{pmatrix})}{\mathbb{E}} \left[ \sigma(s)\sigma(t) \right] + \mathbb{E} \left[ st \right] \right)$$

Convergence in probability follows from the Law of Large Numbers. The righthandside follows from the definition of $\Sigma^l$ and the fact that the output of layer $l-1$ is a from a Gaussian Process. □

### 8.9.2 S-GNN NEURAL TANGENT KERNEL (THEOREM 3.4)

**Theorem 3.4** (Skip-Concatenate GNN NTK). *Having weight dimensions of the hidden layers* $d_{L-1}, d_{L-1}, ..., d_1$ *of a Skip-GNN* $F(X)^L$, *approaching infinity, the NTK is* $\Theta^L \otimes I_{d_l}$

$$\Theta^L = A \left( \Lambda^{L-1} + \left( \dot{\Lambda}^{L-1} \odot \Theta^{L-1} \right) \right) A^T \tag{21}$$

$$\Lambda^L = \sigma_w^2 \frac{1}{2} \left( \mathbb{E} \left[ \sigma(u)\sigma(u^T) \right] + A\Lambda^{L-1}A^T \right) + \sigma_b^2 \tag{22}$$

$$\dot{\Lambda}^L = \sigma_w^2 \frac{1}{2} \left( \mathbb{E} \left[ \dot{\sigma}(u)\dot{\sigma}(u^T) \right] + 1 \right) + \sigma_b^2 \tag{23}$$

*with* $u \sim N(0, A\Lambda^{L-1}A^T)$

$$\Theta^1 = \Lambda^0 = \sigma_w^2 X^T X + \sigma_b^2 \tag{24}$$

*Proof.* Proof is by Induction, similar to the NNGP derivation. The Base Case is just a repetition of 3.2 as $F^1$ has no Skip-Concatenate Connections. We will skip parts which are same as the NTK/ GNTK derivations. Induction Step:

$$\Theta^{l+1} = B + \Gamma$$

$$B = \left( A \left( \frac{\sigma_w^2}{2d_l} G^{l^T} G^l + \sigma_b^2 \mathbb{1}_n \right) A^T \right) \otimes I_{d_{l+1}}$$

$$\frac{\sigma_w^2}{2d_l} G^{l^T} G^l \xrightarrow{d_l \xrightarrow{P} \infty} \sigma_w^2 \begin{pmatrix} \frac{1}{2} \left( \mathbb{E} \left[ \sigma(u_1)\sigma(u_1) \right] + \mathbb{E} \left[ u_1 u_1 \right] \right) & \frac{1}{2} \left( \mathbb{E} \left[ \sigma(u_1)\sigma(u_2) \right] + \mathbb{E} \left[ u_1 u_2 \right] \right) \cdots & \frac{1}{2} \left( \mathbb{E} \left[ \sigma(u_1)\sigma(u_n) \right] + \mathbb{E} \left[ u_1 u_n \right] \right) \\ \vdots & \ddots & \vdots \\ \frac{1}{2} \left( \mathbb{E} \left[ \sigma(u_n)\sigma(u_1) \right] + \mathbb{E} \left[ u_n u_1 \right] \right) & \cdots & \frac{1}{2} \left( \mathbb{E} \left[ \sigma(u_n)\sigma(u_n) \right] + \mathbb{E} \left[ u_n u_n \right] \right) \end{pmatrix}$$

$$\implies B = \Lambda^L \otimes I_{d_{l+1}}$$

$$\Gamma = \frac{\sigma^{2w}}{2d_l} (A \otimes I_{d_l})(I_n \otimes W^{l+1}) \Sigma^l (\Theta^l \otimes I_d) \Sigma^l (I_n \otimes W^{l+1^T})(A^T \otimes I_{d_l}))$$

$$\frac{\sigma_w^2}{2d_l} (I_n \otimes W^{l+1}) \Sigma^l (\Theta^l \otimes I_d) \Sigma^l (I_n \otimes W^{l+1^T})_{IJ_{uv}} = \Theta_{ij} \frac{\sigma_w^2}{2d_l} W_u^{l+1} \Sigma_{II}^l \Sigma_{JJ}^{l^T} (W_v^{l+1})^T$$

$Z := \left( \text{vec}(G^l) \text{vec}(G^l)^T \right)_{IJ}$ and $Z_{11}$ is the upper left block matrix of size $\mathbb{R}^{d_l \times d_l}$.

and $W_u^{l+1} = \left[ \overset{1}{W}_u^{l+1}, \overset{2}{W}_u^{l+1} \right]$ with $\overset{1}{W}_u^{l+1}, \overset{2}{W}_u^{l+1} \in \mathbb{R}^{1 \times d_l}$, so we split $W_u^{l+1}$ into two parts, each of length $d_l$.

$$= \Theta_{ij} \frac{\sigma_w^2}{2d_l} \left( \overset{1}{W}_u^{l+1} Z_{11} \overset{1}{W}_v^{l+1} + \overset{1}{W}_u^{l+1} Z_{12} \overset{2}{W}_v^{l+1} + \overset{2}{W}_u^{l+1} Z_{21} \overset{1}{W}_v^{l+1} + \overset{2}{W}_u^{l+1} Z_{22} \overset{2}{W}_v^{l+1} \right)$$

$$\overset{d_l \overset{P}{\to} \infty}{\longrightarrow} \delta_{uv} \Theta_{ij} \sigma_w^2 \frac{1}{2} \left( \mathbb{E}\left[ \dot{\sigma}(s) \dot{\sigma}(t) \right] + 1 \right)$$

Which follows the same idea for the proof of the NTK. $\qquad\qquad\square$

### 8.10 NTK & GP FOR GRAPH ATTENTION NEURAL NETWORK

The model definition is copied from Section 3.3 for the readers convenience.

**Definition 3.3** (GAT*). *We have two elementwise nonlinearities $\sigma_1$ and $\sigma_1$ which are polynomially bounded, (i.e. $\sigma(x) = c + m|x|$ for some $c, m \in \mathbb{R}_+$) and $c, c_1, c_2$ are defined as above, i.e $c^T = \text{concat}(c_1^T, c_2^T)$.*

$$G^l = \sigma_2(F^l) \text{ and } G^0 = X \text{ with } X \in \mathbb{R}^{d_0 \times n} \tag{28}$$

$$L_{i,j}^{l,h} = A_{ij} \frac{\sigma_c}{\sqrt{2d_{l-1}}} (c_1^{l,h^T} G_{\cdot i}^{l-1} + c_2^{l,h^T} G_{\cdot j}^{l-1}) \text{ with } c_1^{l,h}, c_2^{l,h} \in \mathbb{R}^{d_{l-1}} \tag{29}$$

$$F^{l,h} = G^{l-1} \sigma_1(L^{l,h}) \tag{30}$$

$$F^l = \frac{\sigma_w}{\sqrt{H d_{l-1}}} W^l \left[ F^{l,1}, F^{l,2}, \cdots, F^{l,H} \right]^T \tag{31}$$

*with $W \in \mathbb{R}^{d_l \times H d_{l-1}}$. $F^l$ can also be written as $F^l = \frac{\sigma_w}{\sqrt{H d_{l-1}}} \sum_h^H W^{l,h} F^{l,h}$ and $W^l, c^{l,h} \sim N(0,1)$. The final Network of depth $l$ and heads $H$ is $F^l = \frac{\sigma_w}{\sqrt{H d_{l-1}}} W^l \left[ F^{l,1}, F^{l,2}, \cdots, F^{l,H} \right]^T$*

DERIVATIVES

$$\overset{2}{\Sigma}^l := \frac{\partial \text{vec}(G^l)}{\partial \text{vec}(F^l)} = \text{diag}(\text{vec}(\dot{\sigma}_2(F^l)))$$

$$\frac{\partial \text{vec}(F^l)}{\partial \text{vec}(W^l)} = \frac{\sigma_w}{\sqrt{d_{l-1}}} \left[ F^{l1}, \cdots, F^{lH} \right] \otimes I_{d_l}$$

$$\frac{\partial \text{vec}(F^l)}{\partial \text{vec}(\sigma_1(L^{l,h}))} = I_n \otimes \left( \frac{\sigma_w}{\sqrt{d_{l-1}}} W^{l,h} G^{l-1} \right)$$

$$\overset{1}{\Sigma}^{l,h} := \frac{\partial \text{vec}(\sigma_1(L^{l,h}))}{\partial \text{vec}(L^{l,h})} = \text{diag}(\text{vec}(\dot{\sigma}_1(L^{l,h}))$$

$$\frac{\partial \text{vec}(L^{l,h})}{\partial \text{vec}(c^{l,h})} = \frac{\sigma_c}{\sqrt{2d_{l-1}}} \hat{A} \text{ concat}((\mathbf{1}_n \otimes G^T), (G^T \otimes \mathbf{1}_n)) \text{ with } \hat{A} := \text{diag}(\text{vec}(A^T))$$

derived using $\text{vec}(L^{l,h}) = \frac{\sigma_c}{\sqrt{2d_{l-1}}} \hat{A} \left( (\mathbf{1}_n \otimes G^T) c_1^{l,h} + (G^T \otimes \mathbf{1}_n^T) c_2^{l,h} \right)$

$$\frac{\sigma_c}{\sqrt{2d_{l-1}}}\hat{A}\left((\mathbf{1}_n \otimes G^T c_1^{l,h}) + (G^T c_2^{l,h} \otimes \mathbf{1}_n)\right) = \frac{\sigma_c}{\sqrt{2d_{l-1}}}\hat{A}\operatorname{concat}(\mathbf{1}_n \otimes I_n, I_n \otimes \mathbf{1}_n)\operatorname{concat}(G,G)^T$$

$$\frac{\partial \operatorname{vec}(L^{l,h})}{\partial \operatorname{vec}(G^{h-1})} = \frac{\sigma_c}{\sqrt{2d_{l-1}}}\hat{A}\left((\mathbf{1}_n \otimes (I_n \otimes c_1^{l,h^T})) + (I_n \otimes c_2^{l,h^T}) \otimes \mathbf{1}_n)\right)$$

$$\frac{\partial \operatorname{vec}(F^l)}{\partial \operatorname{vec}(G^{l-1})} = \frac{1}{\sqrt{H}}\sum_h^H\left[\left(I_n \otimes \left(\frac{\sigma_w}{\sqrt{d_{l-1}}}W^{l,h}G^{l-1}\right)\right)\dot{\Sigma}^{l,h}\frac{\partial \operatorname{vec}(L^{l,h})}{\partial \operatorname{vec}(G^{h-1})} + \left(\sigma_1(L^{l,h})^T \otimes \frac{\sigma_w}{\sqrt{d_{l-1}}}W^{l,h}\right)\right]$$

PROOF STRATEGY FOR GAT* GP & NTK

The proof of the GAT* Gaussian Process follows from (Hron et al., 2020, Theorem 1) which is the Scaled Attention Theorem from (Yang, 2019). Theorem 1/ Scaled Attention Theorem is based on (Yang, 2019; 2020, Mastertheorem) which proves that the infinite width Neural Network connections to GP and NTK hold for a variety of architectures. The authors of Hron et al. (2020) generalizes Theorem 1 to a particular scaling, namely the $d^{-\frac{1}{2}}$ scaling for Attention Neural Networks. This is made precise in Theorem 3 from the aforementioned authors. Adapting Theorem 1 is enough to prove the GAT* Gaussian Process derivation because the GAT* Model does not require the $d^{-\frac{1}{2}}$ scaling. Contrary to the previous proofs in this work which derived the infinite Width Limit for $d_{l-1} \to \infty$ layer after layer, the proofs for the NTK and GP for the GAT* are conducted for $\min\{H, d_{l-1}\} \to \infty$ (i.e. the attention heads and weight widths go to infinity simultaneously for each layer). Similar techniques have been used by (de G. Matthews et al., 2018) to demonstrate that infinite width Neural Nets are Gaussian Processes when all widths go to infinity simultaneously. Before we start, we have to show that our GAT* model meets all assumptions from (Hron et al., 2020, Theorem 1/ Theorem 3). We have to make sure that the output of the GAT* model is bounded by a constant which is independent of $H, d_{l-1}$. This will make it possible to use Lemma 32 from Hron et al. (2020). This way, all assumptions of Theorem 3 will be met. To prove boundedness of our GAT* Model it suffices to prove boundedness for the only component that is different to the Attention Neural Network (defined in Hron et al. (2020)) namely $L_{ij}$ from 3.3. Boundedness of the Model then follows by induction and Hölder Inequality (see (Hron et al., 2020, Lemma 32) which is based on de G. Matthews et al. (2018). Boundedness of $L_{ij}$ follows by direct application of (de G. Matthews et al., 2018, Lemma 19). In words: $L_{ij}$ is just an inner product of a constant vector (the data) with a normally distributed vector. This inner product can be bounded by a constant independent of the size of the normally distributed vector. Each entry $L_{ij}$ is just the same vector $c$ which will establish the bound on all of $L$ using the mentioned Lemma 19. The proof of the NTK is conducted using similar techniques to the NTK proof from (Hron et al., 2020).

### 8.10.1 GAUSSIAN PROCESS (THEOREM 3.5)

**Lemma 8.2.** *If the output of a GAT Layer $l-1$ is a GP, so $\operatorname{vec}(F^{l-1}) \sim GP(0, \Omega \otimes I_{d_{l-1}})$ for a fixed $\Omega$ and the $d_{l-1} \to \infty$, then $\operatorname{vec}(L^l) \sim GP(0, \psi(\Omega))$, with*

$$\psi(\Omega) := \mathbb{E}\big[\sigma_1(u)\sigma_1(u^T)\big] \text{ with } u \sim GP\big(0, \sigma_c J_A(\begin{smallmatrix}\Omega & \Omega \\ \Omega & \Omega\end{smallmatrix})J_A^T \otimes I_{d_{l-1}})\big)$$

$$\text{and } J_A := \operatorname{diag}(\operatorname{vec}(A^T)\operatorname{concat}((\mathbf{1}_n \otimes I_n), (I_n \otimes \mathbf{1}_n))$$

*Proof.* We will demonstrate the proof for $d_{l-1} \to \infty$ sequentially for each layer, utilizing Induction to conclude the proof. (The base case follows from using the Definition of $L^1$). This will make the proof much simpler. In the case of $\min\{d_{l-1}\} \to \infty$, (so all Widths going to infinity simultaneously) the proof can be conducted applying (Yang, 2019, Mastertheorem/ Scaled Attention Neural Networks) or (Hron et al., 2020, Theorem 1/ Theorem 3, Part I)). The proof is conducted for $L^{l,h}$ so for every head $h$ with corresponding $c^{l,h}$ but we will omit $h$ and write $c^l$ instead $c^{l,h}$ and $L^l$ instead $L^l$.

$$\mathbb{E}\big[L_{ij}^l\big] = 0$$

$$\mathbb{E}\big[L_{lm}^l L_{st}^l\big] = \frac{\sigma_c^2}{2d_{l-1}} A_{lm} A_{st} \mathbb{E}\big[\big(c_1^{l^T}(G_{\cdot l} + G_{\cdot m}) + c_2^{l^T}(G_{\cdot l} + G_{\cdot m})\big) + \big(c_1^{l^T}(G_{\cdot s} + G_{\cdot t}) c_2^{l^T}(G_{\cdot s} + G_{\cdot t})\big)\big] =$$

$$= \frac{\sigma_c^2}{2d_{l-1}} A_{lm} A_{st} \bigg( \mathbb{E}\big[c_1^{l^T}(G_{\cdot l} + G_{\cdot m}) c_1^{l^T}(G_{\cdot s} + G_{\cdot t})\big] + \mathbb{E}\big[c_1^{l^T}(G_{\cdot l} + G_{\cdot m}) c_2^{l^T}(G_{\cdot s} + G_{\cdot t})\big]$$

$$+ \mathbb{E}\big[c_2^{l^T}(G_{\cdot l} + G_{\cdot m}) c_1^{l^T}(G_{\cdot s} + G_{\cdot t})\big] + \mathbb{E}\big[c_2^{l^T}(G_{\cdot l} + G_{\cdot m}) c_2^{l^T} G_{\cdot l}s + G_{\cdot t})\big]\bigg)$$

$$= \frac{\sigma_c^2}{2d_{l-1}} A_{lm} A_{st} \bigg( \mathbb{E}\big[c_1^{l^T}(G_{\cdot l} + G_{\cdot m}) c_1^{l^T}(G_{\cdot s} + G_{\cdot t})\big] + \mathbb{E}\big[c_2^{l^T}(G_{\cdot l} + G_{\cdot m}) c_2^{l^T} G_{\cdot l}s + G_{\cdot t})\big]\bigg)$$

$$= \frac{\sigma_c^2}{2d_{l-1}} A_{lm} A_{st} \bigg( \mathbb{E}\big[\sum_{uv} c_{1u}^l c_{1v}^l G_{ul} G_{vs}\big] + \mathbb{E}\big[\sum_{uv} c_{1u}^l c_{1v}^l G_{ul} G_{vt}\big] + \mathbb{E}\big[\sum_{uv} c_{1u}^l c_{1v}^l G_{um} G_{vs}\big] + \mathbb{E}\big[\sum_{uv} c_{1u}^l c_{1v}^l G_{um} G_{vt}\big]$$

$$+ \mathbb{E}\big[\sum_{uv} c_{2u}^l c_{2v}^l G_{ul} G_{vs}\big] + \mathbb{E}\big[\sum_{uv} c_{2u}^l c_{2v}^l G_{ul} G_{vt}\big] + \mathbb{E}\big[\sum_{uv} c_{2u}^l c_{2v}^l G_{um} G_{vs}\big] + \mathbb{E}\big[\sum_{uv} c_{2u}^l c_{2v}^l G_{um} G_{vt}\big]\bigg)$$

$$\xrightarrow{\text{Converges in probability for } d_{l-1} \to \infty}$$

$$\sigma_c^2 A_{lm} A_{st} \bigg( \mathbb{E}\big[\sigma_1(u_l)\sigma_1(u_s)\big] + \mathbb{E}\big[\sigma_1(u_l)\sigma_1(u_t)\big] + \mathbb{E}\big[\sigma_1(u_m)\sigma_1(u_s)\big] + \mathbb{E}\big[\sigma_1(u_m)\sigma_1(u_t)\big]\bigg)$$

with $u = \text{vec}(F^{l-1}) \sim GP(0, \Omega \otimes I_{d_{l-1}})$ □

**Theorem 3.5** (GAT* GP).
*If $\min\{H, d_{l-1}\} \to \infty$ then $F^{l+1}$ (defined as above) is a Gaussian Process with $\text{vec}(F^{l+1}) \sim GP\big(0, \Lambda^l \otimes I_{d_{l+1}}\big)$*

$$\Lambda^l := \mathbb{E}\big[\sigma_2(u)\sigma_2(u^T)\big] \text{ with } u \sim GP\big(0, \text{batchm}\big(\sigma_w^2 \Lambda^{l-1}, \psi(\Lambda^{l-1})\big)\big) \tag{32}$$

$$\psi(\Lambda^l) := \mathbb{E}\big[\sigma_1(v)\sigma_1(v^T)\big] \text{ with } v \sim GP\big(0, \sigma_c^2 \gamma_A(\Omega)\big) \tag{33}$$

*and* $\gamma_A(\Omega) := J_A(\begin{smallmatrix} \Omega & \Omega \\ \Omega & \Omega \end{smallmatrix}) J_A^T$ *and* $J_A = \text{diag}(\text{vec}(A)) \text{concat}\,((\mathbf{1}_n \otimes I_n), (I_n \otimes \mathbf{1}_n))$

$$\Lambda^0 = \frac{1}{d_1} X^T X \tag{34}$$

*Proof Sketch.* Theorem 3.1 can be derived by just applying (Hron et al., 2020, Part II) Theorem 1 (which itself is from Yang (2020) or Part II) Theorem 3). Notice that for proving Part II) Theorem 3) at no point is the actual definition of the Attention Neural Networks used, therefore the proves applies to our model without any adaption.

$$\mathbb{E}\big[\text{vec}(F^l)\big] = 0$$

$$\mathbb{E}\big[\text{vec}(F^l)\,\text{vec}(F^l)^T\big] = \mathbb{E}\big[\big(\sigma_1(L^{l,h}) \otimes \frac{\sigma_w}{\sqrt{d_l}} W^{l,h}\big) G^{l-1^T} G^{l-1} \big(\sigma_1(L^{l,h}) \otimes \frac{\sigma_w}{\sqrt{d_l}} W^{l,h}\big)^T\big]$$

for block indices $I, J$

$$\mathbb{E}\big[\text{vec}(F^l)\,\text{vec}(F^l)^T\big]_{IJ} =$$

$$\mathbb{E}\big[\frac{1}{H}\sum_{h,h'}^{H}\sum_{ST} \big(\sigma_1(L^{l,h}) \otimes I_{d_l}\big)_{IS} \big[\big(I_n \otimes \frac{\sigma_w}{\sqrt{d_l}} W^{l,h}\big) G^{l-1^T} G^{l-1} \big(I_{d_l} \otimes \frac{\sigma_w}{\sqrt{d_l}} W^{l,h^T}\big)\big]_{ST} \big(\sigma_1(L^{l,h}) \otimes I_{d_l}\big)_{TJ}\big]$$

$$= \mathbb{E}\Big[\frac{1}{H}\sum_{h,h'}^{H}\sum_{ST}\sigma_1(L^{l,h})_{is}\frac{\sigma_w^2}{d_l}W^{l,h}(G^{l-1^T}G^{l-1})_{ST}W^{l,h'^T}\sigma_1(L^{l,h'})_{tj}\Big]$$

for indices $u, v$

$$\mathbb{E}\big[\text{vec}(F^l)\,\text{vec}(F^l)^T\big]_{IJ_{uv}} = \mathbb{E}\Big[\frac{1}{H}\sum_{h,h'}^{H}\sum_{ST}\sum_{lm}\sigma_1(L^{l,h})_{is}\sigma_1(L^{l,h'})_{tj}\frac{\sigma_w^2}{d_l}W_{ul}^{l,h}W_{vm}^{l,h'}(G^{l-1^T}G^{l-1})_{ST_{lm}}\Big]$$

$$= \frac{1}{H}\sum_{h,h'}^{H}\sum_{ST}\sum_{lm}\sigma_1(L^{l,h})_{is}\sigma_1(L^{l,h'})_{tj}\mathbb{E}\Big[\frac{\sigma_w^2}{d_l}W_{ul}^{l,h}W_{vm}^{l,h'}(G^{l-1^T}G^{l-1})_{ST_{lm}}\Big]$$

Now by a hand wavy argument we first let $d_{l-1}\to\infty$ to show,

$$\mathbb{E}\Big[\frac{\sigma_w^2}{d_l}W_{ul}^{l,h}W_{vm}^{l,h'}(G^{l-1^T}G^{l-1})_{ST_{lm}}\Big] \overset{P}{\to} \sigma_w^2\Lambda_{ST_{lm}}^{l-1}$$

and then $H\to\infty$ and by the Law of Large Numbers

$$\frac{\sigma_w^2}{H}\sum_{h,h'}^{H}\sum_{ST}\sum_{lm}\sigma_1(L^{l,h})_{is}\sigma_1(L^{l,h'})_{tj}\Lambda_{ST_{lm}} \overset{P}{\to} \sigma_w^2\mathbb{E}\Big[\sum_{ST}\sum_{lm}\sigma_1(L^{l,h})_{is}\sigma_1(L^{l,h'})_{tj}\Lambda_{ST_{lm}}\Big]$$

the righthandside is nothing else than

$$\sigma_w^2\mathbb{E}\Big[\sum_{ST}\sum_{lm}\sigma_1(L^{l,h})_{is}\sigma_1(L^{l,h'})_{tj}\Lambda_{ST_{lm}}\Big] = \sigma_w^2\sum_{st}^{n}\mathbb{E}\big[\sigma_1(L^{l,h})_{is}\sigma_1(L^{(l,h')})_{tj}\big]\Lambda_{st}^{l-1} = \sigma_w^2\sum_{st}^{n}\Lambda_{st}^{l-1}\psi(\Lambda^{l-1})_{IJ_{st}}$$

$$\implies \mathbb{E}\big[\text{vec}(F^l)\,\text{vec}(F^l)^T\big] \to \text{batchm}(\sigma_w^2\Lambda^{l-1}, \psi(\Lambda^{l-1}))$$

For a rigourous treatement and the case of $\min\{H, d_{l-1}\}\to\infty$ the proof can be concluded using (Yang, 2019, Mastertheorem) or (Hron et al., 2020, Theorem 3, Part II)). □

### 8.10.2 GAT*GP Neural Tangent Kernel (Theorem 3.6)

**Theorem 3.6** (GAT* NTK). *The NTK (of a GAT* of depth L) is $\Theta^L\otimes I_{d_l}$*

$$\Lambda^l := \mathbb{E}\big[\sigma_2(v)\sigma_2(v^T)\big]), \quad \dot\Lambda^l := \mathbb{E}\big[\dot\sigma_2(v)\dot\sigma_2(v^T)\big] \text{ with } v\sim GP\Big(0, \Lambda^{l-1}\Big) \tag{35}$$

$$\dot\psi(\Lambda^l) := \mathbb{E}\big[\dot\sigma_1(v)\dot\sigma_1(v^T)\big] \text{ with } v\sim GP\Big(0, \sigma_c^2\gamma_A(\Omega)\Big) \tag{36}$$

$$\Theta^1 = \Lambda^0 = \sigma_w^2\frac{1}{d_1}X^TX \tag{37}$$

$$\Theta^L = \text{batchm}\Big[\sigma_w^2\Lambda^{L-1}, \psi(\Lambda^{L-1})\Big] \tag{38}$$

$$+ \text{batchm}\Big[\sigma_w^2\Lambda^{L-1}, \sigma_c^2\Big(\gamma_A(\Lambda^{L-1})\odot\dot\psi(\Lambda^{L-1})\Big)\Big] \tag{39}$$

$$+ \text{batchm}\Big[\sigma_w^2\Lambda^{L-1}, \sigma_c^2\Big(\gamma_A(\Theta^{L-1}\odot\dot\Lambda^{L-1})\odot\dot\psi(\Lambda^{L-1})\Big)\Big] \tag{40}$$

$$+ \text{batchm}\Big[\sigma_w^2\Big(\Theta^{L-1}\odot\dot\Lambda^{L-1}\Big), \psi(\Lambda^{L-1})\Big] \tag{41}$$

*Proof.* As a reminder the NTK of depth $l$ is

$$\frac{\partial\,\text{vec}(F^l(\theta, X))}{\partial\theta}\left(\frac{\partial\,\text{vec}(F^l(\theta, X))}{\partial\theta}\right)^T$$

for parameters $\theta$. We when the width of the hidden layers goes to infinity this expression converges in probability to $\Theta^l \otimes I_{d_l}$. Now to the proof, first realize that:

$$\frac{\partial \operatorname{vec}(F^l(\theta, X))}{\partial \theta} \left( \frac{\partial \operatorname{vec}(F^l(\theta, X))}{\partial \theta} \right)^T = \frac{\partial \operatorname{vec}(F^l)}{\partial \operatorname{vec}(W^l)} \left( \frac{\partial \operatorname{vec}(F^l)}{\partial \operatorname{vec}(W^l)} \right)^T + \frac{1}{H} \sum_h^H \frac{\partial \operatorname{vec}(F^l)}{\partial \operatorname{vec}(c^{l,h})} \left( \frac{\partial \operatorname{vec}(F^l)}{\partial \operatorname{vec}(c^{l,h})} \right)^T$$

$$+ \frac{\partial \operatorname{vec}(F^l)}{\partial \operatorname{vec}(G^{l-1})} \frac{\partial \operatorname{vec}(G^{l-1})}{\partial \operatorname{vec}(F^{l-1})} \frac{\partial \operatorname{vec}(F^{l-1}(\theta, X))}{\partial \theta} \left( \frac{\partial \operatorname{vec}(F^{l-1}(\theta, X))}{\partial \theta} \right)^T \left( \frac{\partial \operatorname{vec}(F^l)}{\partial \operatorname{vec}(G^{l-1})} \frac{\partial \operatorname{vec}(G^{l-1})}{\partial \operatorname{vec}(F^{l-1})} \right)^T$$

Each of the terms is simplified and probability in convergence is proven for $\min\{d_{l-1}, H\} \to \infty$.

**Lemma 8.3.**

$$\frac{\partial \operatorname{vec}(F^l)}{\partial \operatorname{vec}(W^l)} \left( \frac{\partial \operatorname{vec}(F^l)}{\partial \operatorname{vec}(W^l)} \right)^T \xrightarrow{P} \operatorname{batchm} \left( \sigma_w^2 \Lambda^{l-1}, \psi(\Lambda^{l-1}) \right) \otimes I_{d_l}$$

*Proof.*

$$\frac{\partial \operatorname{vec}(F^l)}{\partial \operatorname{vec}(W^l)} \left( \frac{\partial \operatorname{vec}(F^l)}{\partial \operatorname{vec}(W^l)} \right)^T = \frac{\sigma_w^2}{d_{l-1}} \left[ F^{l1}, ..., F^{lH} \right] \left[ F^{l1}, ..., F^{lH} \right]^T \otimes I_{d_l}$$

$$\frac{\sigma_w^2}{H d_{l-1}} \left( \left[ F^{l1}, ..., F^{lH} \right] \left[ F^{l1}, ..., F^{lH} \right]^T \right) = \sigma_w^2 \sum_h^H \frac{1}{H d_{l-1}} \sigma_1(L^{l,h}) G^{l-1^T} G^{l-1} \sigma_1(L^{l,h})$$

We are going to focus on elements $i, j$ and rewrite it

$$\left( \frac{\partial \operatorname{vec}(F^l)}{\partial \operatorname{vec}(W^l)} \left( \frac{\partial \operatorname{vec}(F^l)}{\partial \operatorname{vec}(W^l)} \right)^T \right)_{ij} = \frac{\sigma_w^2}{H} \sum_h^H \sum_{l,s}^n \sigma_1(L^{l,h})_{is} \sigma_1(L^{l,h})_{tj} \frac{\left( G^{l-1^T} G^{l-1} \right)_{st}}{d_{l-1}}$$

$$\xrightarrow{P} \sigma_w^2 \sum_{st}^n \mathbb{E} \left[ \sigma_1(L^{l,h})_{is} \sigma_1(L^{l,h})_{tj} \right] \Lambda_{st}^{l-1} = \sigma_w^2 \sum_{st}^n \Lambda_{st}^{l-1} \psi(\Lambda^{l-1})_{IJ_{st}}$$

To prove convergence in probability (last step). for for $\min\{d_{l-1}, H\} \to \infty$ one can apply (Hron et al., 2020, Lemma 19) or (Yang, 2020, Mastertheorem). □

**Lemma 8.4.**

$$\sum_h^H \frac{\partial \operatorname{vec}(F^l)}{\partial \operatorname{vec}(c^{l,h})} \left( \frac{\partial \operatorname{vec}(F^l)}{\partial \operatorname{vec}(c^{l,h})} \right)^T \xrightarrow{P} \operatorname{batchm} \left[ \sigma_w^2 \Lambda^{l-1}, \sigma_c^2 \left( \gamma_A(\Lambda^{l-1}) \odot \dot{\psi}(\Lambda^{l-1}) \right) \right] \otimes I_{d_l}$$

*Proof.*

$$\frac{1}{H} \sum_h^H \left( \frac{\partial \operatorname{vec}(F^l)}{\partial \operatorname{vec}(\sigma_1(L^{l,h}))} \frac{\partial \operatorname{vec}(\sigma_1(L^{l,h}))}{\partial \operatorname{vec}(L^{l,h})} \frac{\partial \operatorname{vec}(L^{l,h})}{\partial \operatorname{vec}(c^{l,h})} \right) \left( \frac{\partial \operatorname{vec}(F^l)}{\partial \operatorname{vec}(\sigma_1(L^{l,h}))} \frac{\partial \operatorname{vec}(\sigma_1(L^{l,h}))}{\partial \operatorname{vec}(L^{l,h})} \frac{\partial \operatorname{vec}(L^{l,h})}{\partial \operatorname{vec}(c^{l,h})} \right)^T =$$

$$\frac{1}{H} \sum_h^H \left( \left( I_n \otimes \frac{\sigma_w}{\sqrt{d_{l-1}}} W^{l,h} G^l \right) \dot{\Sigma}^{l,h} \frac{\partial \operatorname{vec}(L^{l,h})}{\partial \operatorname{vec}(c^{l,h})} \left( \frac{\partial \operatorname{vec}(L^{l,h})}{\partial \operatorname{vec}(c^{l,h})} \right)^T \dot{\Sigma}^{l,h} \left( I_n \otimes \frac{\sigma_w}{\sqrt{d_{l-1}}} G^{l^T} W^{l,h^T} \right) \right)$$

Focusing on block indices $I, J$

$$\left(\frac{\partial \operatorname{vec}(F^l)}{\partial \operatorname{vec}(c^{l,h})}\left(\frac{\partial \operatorname{vec}(F^l)}{\partial \operatorname{vec}(c^{l,h})}\right)^T\right)_{IJ} = \frac{\sigma_w^2}{Hd_{l-1}}\sum_h^H W^{l,h}G^l \overset{1}{\Sigma}_{II}^{l,h}\left[\frac{\partial \operatorname{vec}(L^{l,h})}{\partial \operatorname{vec}(c^{l,h})}\left(\frac{\partial \operatorname{vec}(L^{l,h})}{\partial \operatorname{vec}(c^{l,h})}\right)^T\right]_{IJ}\overset{1}{\Sigma}_{JJ}^{l,h}G^{l-1^T}W^{l,h^T}$$

Focusing on elements with indices $u, v$ of $I, J$

$$\left(\frac{\partial \operatorname{vec}(F^l)}{\partial \operatorname{vec}(c^{l,h})}\left(\frac{\partial \operatorname{vec}(F^l)}{\partial \operatorname{vec}(c^{l,h})}\right)^T\right)_{IJ_{uv}} = \sigma_w^2\frac{1}{Hd_{l-1}}\sum_h^H\sum_{st}^n\left[\frac{\partial \operatorname{vec}(L^{l,h})}{\partial \operatorname{vec}(c^{l,h})}\left(\frac{\partial \operatorname{vec}(L^{l,h})}{\partial \operatorname{vec}(c^{l,h})}\right)^T\right]_{IJ_{st}}\langle W_u^{l,h},G_{\cdot s}^{l-1}\rangle\langle W_v^{l,h},G_{\cdot t}^{l-1}\rangle\overset{1}{\Sigma}_{II_{ss}}^{l,h}\overset{1}{\Sigma}_{JJ_{tt}}^{l,h}$$

$$= \sigma_w^2\frac{1}{H}\sum_h^H\sum_{st}^n\left[\frac{\partial \operatorname{vec}(L^{l,h})}{\partial \operatorname{vec}(c^{l,h})}\left(\frac{\partial \operatorname{vec}(L^{l,h})}{\partial \operatorname{vec}(c^{l,h})}\right)^T\right]_{IJ_{st}}\overset{1}{\Sigma}_{II_{ss}}^{l,h}\overset{1}{\Sigma}_{JJ_{tt}}^{l,h}\frac{1}{d_{l-1}}\sum_{ab}^{d_{l-1}}W_{ua}^{l,h}W_{vb}^{l,h}G_{as}^{l-1}G_{bt}^{l-1}$$

as a reminder

$$\left[\frac{\partial \operatorname{vec}(L^{l,h})}{\partial \operatorname{vec}(c^{l,h})}\left(\frac{\partial \operatorname{vec}(L^{l,h})}{\partial \operatorname{vec}(c^{l,h})}\right)^T\right]_{IJ_{st}} = \frac{\sigma_c^2}{2d_{l-1}}\langle\hat{A}\operatorname{concat}((\mathbf{1}_n\otimes G^T),(G^T\otimes\mathbf{1}_n))_{I_s},\hat{A}\operatorname{concat}((\mathbf{1}_n\otimes G^T),(G^T\otimes\mathbf{1}_n))_{J_t}\rangle$$

$$= \sigma_c^2\hat{A}_{II_{ss}}\hat{A}_{JJ_{tt}}\frac{\langle G_{\cdot I_s},G_{\cdot J_t}\rangle + \langle G_{\cdot I_s},G_{\cdot J_t}\rangle}{2d_{l-1}}$$

$$= \sigma_w^2\sigma_c^2\frac{1}{H}\sum_h^H\sum_{st}^n\hat{A}_{II_{ss}}\hat{A}_{JJ_{tt}}\frac{\langle G_{\cdot I_s},G_{\cdot J_t}\rangle + \langle G_{\cdot I_s},G_{\cdot J_t}\rangle}{2d_{l-1}}\overset{1}{\Sigma}_{II_{ss}}^{l,h}\overset{1}{\Sigma}_{JJ_{tt}}^{l,h}\frac{1}{d_{l-1}}\sum_{ab}^{d_{l-1}}W_{ua}^{l,h}W_{vb}^{l,h}G_{as}^{l-1}G_{bt}^{l-1}$$

converging in probability for $\min\{d_{l-1},H\}\to\infty$

$$= \delta_{uv}\sigma_w^2\sum_{st}^n(J_A\begin{pmatrix}\Lambda^{l-1}&\Lambda^{l-1}\\\Lambda^{l-1}&\Lambda^{l-1}\end{pmatrix}J_A^T)_{IJ_{st}}\dot{\psi}(\Lambda^{l-1})_{IJ_{st}}\dot{\Lambda}_{st}^{l-1}$$

using our previously defined shorthand definition $\gamma_A(\Omega) := J_A(\begin{smallmatrix}\Omega&\Omega\\\Omega&\Omega\end{smallmatrix})J_A^T$ we end up with

$$= \delta_{uv}\sigma_w^2\sum_{st}^n\gamma_A(\Lambda^{l-1})_{IJ_{st}}\dot{\psi}(\Lambda^{l-1})_{IJ_{st}}\Lambda_{st}^{l-1}$$

Similar to the previous lemma to conclude convergence in probability for $\min\{d_{l-1},H\}\to\infty$ one can use (Hron et al., 2020, Lemma 21) or (Yang, 2020, Master Theorem). $\square$

**Lemma 8.5.**

$$\frac{\partial \operatorname{vec}(F^l)}{\partial \operatorname{vec}(G^{l-1})}\frac{\partial \operatorname{vec}(G^{l-1})}{\partial \operatorname{vec}(F^{l-1})}\left(\Theta^{l-1}\otimes I_{d_{l-1}}\right)\left(\frac{\partial \operatorname{vec}(F^l)}{\partial \operatorname{vec}(G^{l-1})}\frac{\partial \operatorname{vec}(G^{l-1})}{\partial \operatorname{vec}(F^{l-1})}\right)^T$$

$$\xrightarrow{P}$$

$$\operatorname{batchm}\left[\sigma_w^2\Lambda^{l-1},\sigma_c^2\left(\gamma_A(\Theta^{l-1}\odot\dot{\Lambda}^{l-1})\odot\dot{\psi}(\Lambda^{l-1})\right)\right] + \operatorname{batchm}\left[\sigma_w^2\left(\Theta^{l-1}\odot\dot{\Lambda}^{l-1}\right),\psi(\Lambda^{l-1})\right]$$

*Proof.*

$$\left(\frac{\partial \operatorname{vec}(F^l)}{\partial \operatorname{vec}(G^{l-1})}\frac{\partial \operatorname{vec}(G^{l-1})}{\partial \operatorname{vec}(F^{l-1})}\right)\left(\Theta^{l-1}\otimes I_{d_{l-1}}\right)\left(\frac{\partial \operatorname{vec}(F^l)}{\partial \operatorname{vec}(G^{l-1})}\frac{\partial \operatorname{vec}(G^{l-1})}{\partial \operatorname{vec}(F^{l-1})}\right)^T =$$

$$\frac{\partial \operatorname{vec}(F^l)}{\partial \operatorname{vec}(G^{l-1})} \hat{\Theta}^l \frac{\partial \operatorname{vec}(F^l)}{\partial \operatorname{vec}(G^{l-1})}^T$$

$\hat{\Theta}^l_{IJ} := \Theta^{l-1}_{ij} \overset{1}{\Sigma}^{l,h}_{II} \overset{1}{\Sigma}^{l,h'}_{JJ}$, similar to (Hron et al., 2020, Section B2.2 Indirect Contributions). Also note that for $d_{l-1} \to \infty$, $\hat{\Theta}^l \overset{P}{\to} (\Theta^{l-1} \odot \dot{\Lambda}^{l-1}) \otimes I_{d_{l-1}}$. (see for example (Yang, 2019, Master Theorem)). Now continuing with the simplification.

$$\frac{1}{H} \sum_{h,h'}^{H} \left[ \left( I_n \otimes \left( \frac{\sigma_w}{\sqrt{d_l}} W^{l,h} G^{l-1} \right) \right) \overset{1}{\Sigma}^{l,h} \frac{\partial \operatorname{vec}(L^{l,h})}{\partial \operatorname{vec}(G^{h-1})} + \left( \sigma_1(L^{l,h}) \otimes \frac{\sigma_w}{\sqrt{d_l}} W^{l,h} \right) \right]$$

$$\cdot \hat{\Theta}^l \left[ \left( I_n \otimes \left( \frac{\sigma_w}{\sqrt{d_l}} W^{l,h'} G^{l-1} \right) \right) \overset{1}{\Sigma}^{l,h'} \frac{\partial \operatorname{vec}(L^{l,h})}{\partial \operatorname{vec}(G^{h-1})} + \left( \sigma_1(L^{l,h'}) \otimes \frac{\sigma_w}{\sqrt{d_{l-1}}} W^{l,h'} \right) \right]^T$$

$$= \frac{1}{H} \sum_{h,h'}^{H} \left( A^{l,h,h'} + B^{l,h,h'} + C^{l,h,h'} + C^{l,h,h'^T} \right)$$

Now we will simplify this final expression using the following lemmas to conclude this proof. $\qquad \square$

**Lemma 8.6.**

$$\frac{1}{H} \sum_{h,h'}^{H} A^{l,h,h'} \overset{P}{\to} \operatorname{batchm} \left[ \sigma_w^2 \Lambda^{l-1}, \sigma_c^2 \left( \gamma_A (\Theta^{l-1} \odot \dot{\Lambda}^{l-1}) \odot \dot{\psi}(\Lambda^{l-1}) \right) \right] \otimes I_{d_l}$$

*Proof.*

$$A^{l,h,h'} = \left[ \left( I_n \otimes \left( \frac{\sigma_w}{\sqrt{d_l}} W^{l,h} G^{l-1} \right) \right) \overset{1}{\Sigma}^{l,h} \frac{\partial \operatorname{vec}(L^{l,h})}{\partial \operatorname{vec}(G^{h-1})} \right] \hat{\Theta}^l \left[ \left( I_n \otimes \left( \frac{\sigma_w}{\sqrt{d_{l-1}}} W^{l,h'} G^{l-1} \right) \right) \overset{1}{\Sigma}^{l,h'} \frac{\partial \operatorname{vec}(L^{l,h})}{\partial \operatorname{vec}(G^{h-1})} \right]^T$$

$$= \left( I_n \otimes \left( \frac{\sigma_w}{\sqrt{d_l}} W^{l,h} G^{l-1} \right) \right) \overset{1}{\Sigma}^{l,h} \frac{\partial \operatorname{vec}(L^{l,h})}{\partial \operatorname{vec}(G^{h-1})} \hat{\Theta}^l \left( \frac{\partial \operatorname{vec}(L^{l,h})}{\partial \operatorname{vec}(G^{h-1})} \right)^T \overset{1}{\Sigma}^{l,h'} \left( I_n \otimes \left( \frac{\sigma_w}{\sqrt{d_{l-1}}} G^{l-1^T} W^{l,h'^T} \right) \right)$$

focusing on block indices $I, J$

$$A^{l,h,h'}_{IJ} = \frac{\sigma_w^2}{d_{l-1}} W^{l,h} G^{l-1} \overset{1}{\Sigma}^{l,h}_{II} \left[ \frac{\partial \operatorname{vec}(L^{l,h})}{\partial \operatorname{vec}(G^{h-1})} \hat{\Theta}^l \left( \frac{\partial \operatorname{vec}(L^{l,h})}{\partial \operatorname{vec}(G^{h-1})} \right)^T \right]_{IJ} \overset{1}{\Sigma}^{l,h'}_{JJ} G^{l-1^T} W^{l,h'^T}$$

$$\left[ \frac{\partial \operatorname{vec}(L^{l,h})}{\partial \operatorname{vec}(G^{h-1})} \hat{\Theta}^l \left( \frac{\partial \operatorname{vec}(L^{l,h})}{\partial \operatorname{vec}(G^{h-1})} \right)^T \right]_{IJ_{st}} = \frac{\sigma_c^2}{2d_{l-1}} \tilde{A}_{II_{ss}} \tilde{A}_{JJ_{tt}} \left( c_1^T \hat{\Theta}^l_{st} c_1 + c_1^T \hat{\Theta}^l_{sJ} c_2 + c_2^T \hat{\Theta}^l_{It} c_1 + c_2^T \hat{\Theta}^l_{IJ} c_2 \right)$$

with $\hat{\Theta}^l_{st} \in \mathbb{R}^{d_{l-1} \times d_{l-1}}$. Focusing on element indices $u, v$

$$A^{l,h,h'}_{IJ_{uv}} = \frac{\sigma_w^2}{d_{l-1}} \sum_{st} \left[ \frac{\partial \operatorname{vec}(L^{l,h})}{\partial \operatorname{vec}(G^{h-1})} \hat{\Theta}^l \left( \frac{\partial \operatorname{vec}(L^{l,h})}{\partial \operatorname{vec}(G^{h-1})} \right)^T \right]_{IJ_{st}} \overset{1}{\Sigma}^{l,h}_{II_{ss}} \overset{1}{\Sigma}^{l,h'}_{JJ_{tt}} \langle W_u, G^{l-1}_{\cdot s} \rangle \langle W_v, G^{l-1}_{\cdot t} \rangle$$

$$A^{l,h,h'}_{IJ_{uv}} = \frac{\sigma_w^2 \sigma_c^2}{2d_{l-1}^2} \sum_{st} \tilde{A}_{II_{ss}} \tilde{A}_{JJ_{tt}} \left( c_1^T \hat{\Theta}^l_{st} c_1 + c_1^T \hat{\Theta}^l_{sJ} c_2 + c_2^T \hat{\Theta}^l_{It} c_1 + c_2^T \hat{\Theta}^l_{IJ} c_2 \right) \overset{1}{\Sigma}^{l,h}_{II_{ss}} \overset{1}{\Sigma}^{l,h'}_{JJ_{tt}} \langle W_u, G^{l-1}_{\cdot s} \rangle \langle W_v, G^{l-1}_{\cdot t} \rangle$$

follows the previous ideas and the the fact that $\hat{\Theta}^l \xrightarrow{P} (\Theta^{l-1} \odot \dot{\Lambda}^{l-1}) \otimes I_{d_{l-1}}$ as in 8.5

Convergencence in probability for $\min\{d_{l-1}, H\} \to \infty$

$$\frac{1}{H} \sum_{h,h'} A^{l,h,h'}_{IJ_{uv}} \xrightarrow{P} \delta_{uv} \delta_{hh'} \sum_{st} \tilde{A}_{II_{ss}} \tilde{A}_{JJ_{tt}} \dot{\Lambda}^{l-1}_{IJ_{st}} \Theta^{l-1}_{st} \Theta^{l-1}_{ij} \Lambda^{l-1}_{st}$$

which can be conducted using (Yang, 2020, Master Theorem) or (Hron et al., 2020, Lemma 23). $\qquad\square$

**Lemma 8.7.**

$$\frac{1}{H} \sum_{h,h'}^{H} B^{l,h,h'} \xrightarrow{P} \text{batchm} \left[ \sigma_w^2 \left( \Theta^{l-1} \odot \dot{\Lambda}^{l-1} \right), \psi(\Lambda^{l-1}) \right] \otimes I_{d_{l-1}}$$

*Proof.*

$$B^{l,h,h'} = \left( \sigma_1(L^{l,h}) \otimes \frac{\sigma_w}{\sqrt{d_l}} W^{l,h} \right) \hat{\Theta}^l \left( \sigma_1(L^{l,h'}) \otimes \frac{\sigma_w}{\sqrt{d_l}} W^{l,h'} \right)^T$$

$$B^{l,h,h'}_{IJ} = \sum_{ST} \left( \sigma_1(L^{l,h}) \otimes I_{d_l} \right)_{IS} \left[ \left( I_n \otimes \frac{\sigma_w}{\sqrt{d_l}} W^{l,h} \right) \hat{\Theta}^l \left( I_{d_n} \otimes \frac{\sigma_w}{\sqrt{d_l}} W^{l,h'^T} \right) \right]_{ST} \left( \sigma_1(L^{l,h'}) \otimes I_{d_l} \right)_{TJ}$$

$$B^{l,h,h'}_{IJ} = \sum_{ST} \sigma_1(L^{l,h})_{is} \frac{\sigma_w^2}{d_l} W^{l,h'} \hat{\Theta}^l_{ST} W^{l,h'^T} \sigma_1(L^{l,h'})_{tj}$$

finally proof in convergence

$$\frac{1}{H} \sum_{h}^{H} B^{l,h,h'}_{IJ} \xrightarrow{P} \sum_{st}^{n} \mathbb{E} \left[ \sigma_1(L^{l,h})_{is} \sigma_1(L^{l,h'})_{tj} \right] \Theta^{l-1}_{st} \dot{\Lambda}^{l-1}_{st} = \sum_{st}^{n} \psi(\Lambda^{l-1})_{IJ_{st}} \Theta^{l-1}_{st} \dot{\Lambda}^{l-1}_{st}$$

To proof convergence in probability for $\min\{d_{l-1}, H\} \to \infty$ on can use (Yang, 2020, Master Theorem) or (Hron et al., 2020, Lemma 22). $\qquad\square$

**Lemma 8.8.**

$$\frac{1}{H} \sum_{h,h'}^{H} C^{l,h,h'} \xrightarrow{P} 0$$

*Proof.*

$$\frac{1}{H} \sum_{h,h'}^{H} C^{l,h,h'} = \frac{1}{H} \sum_{h,h'}^{H} \left( I_n \otimes \left( \frac{\sigma_w}{\sqrt{2d_l}} W^{l,h} G^{l-1} \right) \right) \dot{\Sigma}^{l,h} \frac{\partial \text{vec}(L^{l,h})}{\partial \text{vec}(G^{h-1})} \hat{\Theta}^l \left( \sigma_1(L^{l,h'}) \otimes \frac{\sigma_w}{\sqrt{d_{l-1}}} W^{l,h'} \right)^T$$

focusing on block indices $IJ$

$$\frac{1}{H} \sum_{h,h'}^{H} C^{l,h,h'}_{IJ} = \frac{1}{H} \sum_{h,h'}^{H} \frac{\sigma_w}{\sqrt{2d_l}} W^{l,h} G^{l-1} \dot{\Sigma}^{l,h}_{II} \left( \frac{\partial \text{vec}(L^{l,h})}{\partial \text{vec}(G^{h-1})} \hat{\Theta}^l \right)_I \left( (\sigma_1(L^{l,h'})_j)^T \otimes \frac{\sigma_w}{\sqrt{d_{l-1}}} W^{l,h'^T} \right)$$

focusing on indices $uv$

$$\frac{1}{H} \sum_{h,h'}^{H} C^{l,h,h'}_{IJ_{uv}} = \frac{\sigma_w^2 \sigma_c}{\sqrt{2} H d_{l-1}^{\frac{3}{2}}} \sum_{h,h'}^{H} \sum_{st} W^{l,h}_u G^{l-1}_{\cdot s} \dot{\Sigma}^{l,h}_{II_{ss}} \tilde{A}_{II_{ss}} (c_1^T \hat{\Theta}^l_s + c_2^T \hat{\Theta}^l_I) \left( (\sigma_1(L^{l,h'})_j)^T \otimes \frac{\sigma_w}{\sqrt{d_{l-1}}} (W^{l,h'}_v)^T \right)$$

$$= \frac{\sigma_w^2 \sigma_c}{\sqrt{2} H d_{l-1}^{\frac{3}{2}}} \sum_{h,h'}^{H} \sum_{stz} \sigma_1(L^{l,h'})_{[j,\lfloor t \bmod n \rfloor+1]} W_{uz}^{l,h} G_{zs}^{l-1} \overset{1}{\Sigma}_{II_{ss}}^{l,h} \tilde{A}_{II_{ss}} (c_1^T \hat{\Theta}_{st}^l + c_2^T \hat{\Theta}_{It}^l) W_v^{l,h'^T}$$

Define $\nu(t) := \lfloor t \bmod n \rfloor + 1$

$$\frac{\sigma_w^2 \sigma_c}{\sqrt{2} H d_{l-1}^{\frac{3}{2}}} \sum_{h,h'}^{H} \sum_{stzlm} \sigma_1(L^{l,h'})_{j\nu(j)} W_{uz}^{l,h} G_{zs}^{l-1} \overset{1}{\Sigma}_{II_{ss}}^{l,h} \tilde{A}_{II_{ss}} \left( c_{1l} W_{vm}^{l,h'} \hat{\Theta}_{st_{lm}}^l + c_{2l} W_{vm}^{l,h'} \hat{\Theta}_{It_{lm}}^l \right)$$

Using Chebyshevs Inequality

$$\mathbb{P}\left( |S - \mathbb{E}S| \geq \delta \right) \leq \frac{\mathbb{E}[S^2] - \mathbb{E}[S]^2}{\delta^2}$$

we show that $\mathbb{E}[S^2] = \mathbb{E}[S^2]$ (and $\mathbb{E}[S] = 0$) for $\min\{H, d_{l-1}\} \to \infty$ to finish the proof.
The cross terms where Expectation is taken of $c_1$ and $c_2$ are zero.
Defining $\Gamma_{h_1 h_2 h_1' h_2' s_1 s_2} := \sigma_1(L^{l,h_1'})_{j\nu(t_1)} \sigma_1(L^{l,h_2'})_{j\nu(t_2)} \overset{1}{\Sigma}_{II_{s_1 s_1}}^{l,h_1} \overset{1}{\Sigma}_{II_{s_2 s_2}}^{l,h_2} (L^{l,h_2'})_{j\nu(t_2)} \tilde{A}_{II_{s_1 s_1}} \tilde{A}_{II_{s_2 s_2}}$

$$\mathbb{E}\left[ \left( \frac{1}{H} \sum_{h,h'}^{H} C_{IJ_{uv}}^{l,h,h'} \right)^2 \right] =$$

$$\frac{\sigma_w^4 \sigma_c^2}{2H^2 d_{l-1}^3} \left( \mathbb{E}\left[ \sum_{\substack{h_1 h_2 \\ h_1' h_2' \\ s_1 s_2 \\ t_1 t_2 \\ z_1 z_2 \\ l_1 l_2 \\ m_1 m_2}} \Gamma_{h_1 h_2 h_1' h_2' s_1 s_2} W_{uz_1}^{l,h_1} G_{z_1 s_1}^{l-1} W_{uz_2}^{l,h_2} G_{z_2 s_2}^{l-1} c_{1l_1} W_{vm_1}^{l,h_1'} \hat{\Theta}_{s_1 t_{1_{l_1 m_1}}}^l c_{1l_2} W_{vm_2}^{l,h_2'} \hat{\Theta}_{s_2 t_{2_{l_2 m_2}}}^l \right] \right.$$

$$\left. + \mathbb{E}\left[ \sum_{\substack{h_1 h_2 \\ h_1' h_2' \\ s_1 s_2 \\ t_1 t_2 \\ z_1 z_2 \\ l_1 l_2 \\ m_1 m_2}} \Gamma_{h_1 h_2 h_1' h_2' s_1 s_2} W_{uz_1}^{l,h_1} G_{z_1 s_1}^{l-1} W_{uz_2}^{l,h_2} G_{z_2 s_2}^{l-1} c_{2l_1} W_{vm_1}^{l,h_1'} \hat{\Theta}_{It_{1_{l_1 m_1}}}^l c_{2l_2} W_{vm_2}^{l,h_2'} \hat{\Theta}_{It_{2_{l_2 m_2}}}^l \right] \right)$$

By the boundedness of all components of $\Gamma$ (boundedness in the sense that for growing $\min\{H, d_{l-1}\} \to \infty$ the expression is bounded by a constant not depending on $d_{l-1}$ or $H$) and Hölderlins Inequality it follows that $\Gamma$ is bounded by a constant only depending polynomially on X.

$$\frac{\zeta \sigma_w^4 \sigma_c^2}{2H^2 d_{l-1}^3} \mathbb{E}\left[ \sum_{\substack{h_1 h_2 \\ h_1' h_2' \\ s_1 s_2 \\ t_1 t_2 \\ z_1 z_2 \\ l_1 l_2 \\ m_1 m_2}} W_{uz_1}^{l,h_1} G_{z_1 s_1}^{l-1} W_{uz_2}^{l,h_2} G_{z_2 s_2}^{l-1} c_{1l_1} W_{vm_1}^{l,h_1'} \hat{\Theta}_{s_1 t_{1_{l_1 m_1}}}^l c_{1l_2} W_{vm_2}^{l,h_2'} \hat{\Theta}_{s_2 t_{2_{l_2 m_2}}}^l \right]$$

Note that for $\min\{d_{l-1}, H\} \to \infty$, $\hat{\Theta}^l_{IJ_{lm}} \xrightarrow{P} 0$ for $l \neq m$ (from Lemma 8.5). Therefore

$$\frac{\zeta \sigma_w^4 \sigma_c^2}{2H^2 d_{l-1}^3} \mathbb{E}\left[\sum_{\substack{h_1 h_2 \\ h_1' h_2' \\ s_1 s_2 \\ t_1 t_2 \\ z_1 z_2 \\ l \\ m}} W^{l,h_1}_{uz_1} G^{l-1}_{z_1 s_1} W^{l,h_2}_{uz_2} G^{l-1}_{z_2 s_2} c_{1l} W^{l,h_1'}_{vl} \hat{\Theta}^l_{s_1 t_{1_{ll}}} c_{1m} W^{l,h_2'}_{vm} \hat{\Theta}^l_{s_2 t_{2_{mm}}}\right]$$

The case $u \neq v$

$$\frac{\zeta \sigma_w^4 \sigma_c^2}{2H^2 d_{l-1}^3} \mathbb{E}\left[\sum_{\substack{h \\ h' \\ s_1 s_2 \\ t_1 t_2 \\ z \\ l}} W^{l,h}_{uz} G^{l-1}_{zs_1} W^{l,h}_{uz} G^{l-1}_{zs_2} c_{1l} c_{1l} W^{l,h'}_{vl} W^{l,h'}_{vl} \hat{\Theta}^l_{s_1 t_{1_{ll}}} \hat{\Theta}^l_{s_2 t_{2_{ll}}}\right] \overset{\min\{d_{l-1}, H\} \to \infty}{=} 0$$

For $\min\{d_{l-1}, H\} \to \infty$, $h, h', z, l$ are the indices that go to infinity, but $H^2 d_{l-1}^3$ is growing faster than $h, h', z, l$ and therefore the expression is zero. The argument can be repeated for the righthandside to conclude for the case $u \neq v$.

The case $u = v$

$$\frac{\zeta \sigma_w^4 \sigma_c^2}{2H^2 d_{l-1}^3} \mathbb{E}\left[\sum_{\substack{h_1 h_2 \\ h_1' h_2' \\ s_1 s_2 \\ t_1 t_2 \\ z_1 z_2 \\ l \\ m}} W^{l,h_1}_{uz_1} G^{l-1}_{z_1 s_1} W^{l,h_2}_{uz_2} G^{l-1}_{z_2 s_2} c_{1l} W^{l,h_1'}_{ul} \hat{\Theta}^l_{s_1 t_{1_{ll}}} c_{1m} W^{l,h_2'}_{um} \hat{\Theta}^l_{s_2 t_{2_{mm}}}\right]$$

$$= \frac{\zeta \sigma_w^4 \sigma_c^2}{2H^2 d_{l-1}^3} \mathbb{E}\left[\sum_{\substack{h_1 h_2 \\ h_1' h_2' \\ s_1 s_2 \\ t_1 t_2 \\ z_1 z_2 \\ l \\ m}} W^{l,h_1'}_{ul} W^{l,h_2'}_{um} W^{l,h_1}_{uz_1} W^{l,h_2}_{uz_2} c_{1l} c_{1m} G^{l-1}_{z_1 s_1} G^{l-1}_{z_2 s_2} \hat{\Theta}^l_{s_1 t_{1_{ll}}} \hat{\Theta}^l_{s_2 t_{2_{mm}}}\right]$$

The sum over $h_1, h_2, h_1', h_2'$ can be split over:

1. $h_1' = h_2'$ and $h_1 = h_2$ with $l = m$ and $z_1 = z_2$

2. $h_1' = h_1$ and $h_2' = h_2$ with $l = z_1$ and $m = z_2$

3. $h_1' = h_2$ and $h_2' = h_1$ with $l = z_2$ and $m = z_1$

With each case being a sum of which four indices grow to infinity. And therefore again the same argument holds. $\qquad \square$

**Corollary 8.2.** *The NTK (of a GAT\* of depth $L$ with $\sigma_1$ being the Identitiy Function ) is $\Theta^L \otimes I_{d_l}$, with*

$$\Lambda^l := \mathbb{E}\left[\sigma_2(v)\sigma_2(v^T)\right], \ \dot{\Lambda}^l := \mathbb{E}\left[\dot{\sigma}_2(v)\dot{\sigma}_2(v^T)\right] \ with \ v \sim GP\left(0, \Lambda^{l-1}\right)$$

$$\Theta^1 = \Lambda^0 = \sigma_w^2 \frac{1}{d_1} X^T X$$

$$\Theta^L = 2 \cdot \text{batchm}\left[\sigma_w^2 \Lambda^{L-1}, \sigma_c^2 \gamma_A(\Lambda^{L-1})\right] + \text{batchm}\left[\sigma_w^2 \Lambda^{L-1}, \sigma_c^2 \gamma_A\left(\Theta^{L-1} \odot \dot{\Lambda}^{L-1}\right)\right]$$

$$+ \text{batchm}\left[\sigma_w^2 \left(\Theta^{L-1} \odot \dot{\Lambda}^{L-1}\right), \sigma_c^2 \gamma_A(\Lambda^{L-1})\right]$$

