# OpenReview forum: "Graph Neural Tangent Kernel and Graph Neural Network Gaussian Processes for Node Classification/ Regression"
_ICLR.cc/2024/Conference — Submitted to ICLR 2024_

### Official Review · Reviewer_UHJu · 2023-10-24

**Soundness:** 2 fair
**Presentation:** 3 good
**Contribution:** 2 fair
**Rating:** 5
**Confidence:** 4

**Summary:**

This paper presents explicit formulas for computing NNGP and NTK for some GNN variants including those with skip-concatenation and attention mechanism in node classification/regression tasks. Experimental results show kernel ridge regression w.r.t. these kernels has competitive performance on some node classification and regression tasks.

**Strengths:**

1. The closed form expression for NTK and NNGP for GNNs with skip-concatenation and attentions seems interesting and potentially useful for understanding the efficacy of different GNN architectures.
2. The structure of the paper is clear and easy to follow.

**Weaknesses:**

1. The novelty and technical contribution is limited. GNTK has been studied in graph-level tasks, and also extended to node-level ones in some published papers that are neglected in this work, e.g. [1-2] (though they did not explicitly claim that as a contribution). While I understand considering other architectures might be novel, regrettably there is no discussions on how they might help theoretical understandings or become practically useful (incorporating which can greatly strengthen the paper).
2. Theorem 3.2 seems incorrect. $A^\top$ should be outside the bracket in $\mathbf \Theta$ otherwise the kernel matrix is not symmetric and thus invalid. (This could be a typo rather than technical flaw.)
3. The method seems not scalable or efficient, though it is a common limitation for applying NTKs of other types of neural networks. PS, the datasets in this paper are not “large” (or even “medium”) as has been claimed in 4.1. For large datasets, the authors can consider those in [3].
4. For experiments, the dataset split is not standard, and the standard deviation is not reported. Whether the results are statistically significant is unknown.
5. Many formatting issues and nonstandard notations impair the readability. To name a few: abuse of capital letters in writing; unnumbered equations; wrap lines in theorems (3.2, 3.3, 3.5); precision is not consistent for Crocodile dataset. Many expressions throughout the paper are also informal for an academic paper.

[1] Graph Neural Networks are Inherently Good Generalizers: Insights by Bridging GNNs and MLPs, in ICLR 2023
[2] Graph Neural Tangent Kernel: Convergence on Large Graphs, in ICML 2023
[3] Open Graph Benchmark: Datasets for Machine Learning on Graphs, in NeurIPS 2020

**Questions:**

1. Are there any theoretical insights we can draw from the explicit formula of NTK for GNNs with skip-concatenation and attentions about, e.g. why these architectures work?
2. Why GAT\*GP and GAT\*NTK perform worse than other kernels?
3. How exactly is the efficiency of KRR-based methods compared with standard GNNs?

---

> ### Author Response · Authors · 2023-11-15
> **Addressing Review**
>
> We argue that the main contributions are novel starting with adressing weakness 1).
>
> ------Weakness 1)-----
>
> The GNTK formula in [1] is based on the paper "Graph Neural Tangent Kernel: Fusing Graph Neural Networks with Graph Kernels".
> [1] is explicitly calculating a two layer GNTK closed form expression in Lemma 2. We compare their expression to ours. Their expression:
>
> Lemma 2: The explicit form of GNTK feature map for a two-layer GNN $\phi_{gnn}(x)$ with average aggregation and ReLU activation in node regression is:
> $$
> \phi_{gnn}(x_i) =c \sum_{j \in \mathcal N_i  \cup\{i\}}  \left[\mathbf{X}^{\top} a_j     \cdot \mathbb{I_{+}}    \left(w^{(k)^{\top}} \mathbf{X}^{\top} a_j \right), w^{(k)^{\top}} \mathbf{X}^{\top} a_j \cdot \mathbb{I_{+}}\left(w^{(k)^{\top}} \mathbf{X}^{\top} a_j \right), \ldots\right]
> $$
> where $c=O(\tilde{d}^{-1})$ is a constant proportional to the inverse of node degree, $\mathbf X\in \mathbb R^{n\times d}$ is node features, $a_i\in \mathbb R^{n}$ denotes adjacency vector of node $i$, $w^{(k)} \sim \mathcal{N}(\mathbf{0}, I_d)$ is a random Gaussian vector in $\mathbb R^d$, two components in the brackets repeat infinitely many times with $k$ ranging from $1$ to $\infty$, and $\mathbb{I}_{+}$ is an indicator function that outputs $1$ if the input is positive otherwise $0$.
>
> Their NTK is $\Theta_{i,j} = NTK(x_{i}, x_{j}) = \phi_{gnn}(x_i)^{T} \phi_{gnn}(x_j)$
>
> Now our NTK expression:
> $$
> \Theta^{L} =  A \left( \Lambda^{L-1} +  \left( \dot{\Lambda}^{L-1}  \odot \Theta^{L-1} \right) \right)  A^{T}
> $$
> $$
> \Lambda^{L} = \sigma_{w}^{2} \mathbb{E} \left[ \sigma(u) \sigma(u^{T}) \right] +\sigma_{b}^{2}
> $$
> $$
> \dot{\Lambda}^{L} = \sigma_{w}^{2}\mathbb{E} \left[ \dot{\sigma}(u) \dot{\sigma}(u^{T}) \right] +\sigma_{b}^{2} \text{ with } u \sim N(0, A\Lambda^{L-1}A^{T})
> $$
> $$
>  \Theta^{1} = \Lambda^{0} = \sigma_{w}^{2} X^{T}X + \sigma_{b}^{2}
> $$
>
> Both have to be the same but our derivation is much cleaner and simpler to comprehend, dervied differently and we can make statements about psd of the kernel.
> Proof of Lemma 2 has to take into account that it is a **two layer** GNN whereas our formula is recursive and for any number of layers.
>
> Paper [1] requires the GNTK only, to prove some of their claims. They don't do extensive evaluation on any datasets.
>
> The paper [2] Graph Neural Tangent Kernel: Convergence on Large Graphs is only concerned with the empirical NTK and **not** with the infinite width limit closed form expression.
>
> Their are no simulations (on NTK Theory) on any paper mentioned, or to the best of our knowledge any paper existing, validating NTK Theory. We included simulations.
>
> We added a Discussion Section explaining how these closed form expression might help in the future. Theoretical section at the end of GNN NTK proof, comments on the positive definiteness and the effect of the Graph Scaling to the Kernel matrix and its theoreticl implications.
>
> -----Weakness 2-----------
>
> Was a typo is fixed
>
> -------------Weakness 3---------
>
> We removed any statements about the size of the dataset from the paper as your remark is correct.
>
> ----------------Weakness 4-----------
>
> The dataset splits on the classification tasks are very common. Cora has fixed test/train/split due to unlabelled nodes. The other splits were exactly modelled after the papers: Semi-Supervised Classification with Graph Convolutional Networks, Graph Attention Networks, Graph Neural Network-Inspired Kernels for Gaussian Processes in Semi-Supervised Learning.
> The Kernels and Gaussian Processes do not have any random weights and for variance on the weights one can consider the original papers were the Neural Network counterparts were evaluated.
>
> ----Weakness 5
>
> Changed most to small letters (Neural Net/ Graph Neural Net is left with capital letters). Added numbers in equation in the main part of the paper. The GAT*NTK spans 4 lines for readability. It is impossible to put it in one line. We tried reformulated large parts improving readability and using more scientific language.
>
> Question 1
>
> Theoretical insights addressed in GNN NTK section: show how graph "normalization" affects positive definitess of kernel matrix and draw multiple connection to more theoretical possible future research directions.
> Other empirical insights are Role $\sigma_{b}^{2}$ on Regression and Experiments showing Overparametrization leads to Overfitting in GNNs.
>
> Question 2
>
> Don't know currently why GAT performs worse on Regression.
>
> Question 3
>
> Added runtime and memory analysis in Appendix.
>
> We added to the paper:
> - Wiki Dataset  + Experiments
> - Different Nonlinearities for GAT
> - Experiment on Different Widths for GNNs
> - Runtime Analysis & Memory Analysis
> - Discussion Section
> - Theoretical Insights (psd of Kernel Matrix after GNTK proof)
> - Fixed Bug on GAT*NTK/GP
>
> The review substantially improved the paper. Many things were revised and we would nicely ask the reviewer to reread the revised version of the paper and reconsider their evaluation. Thanks!

---

> > ### Comment · Reviewer_UHJu · 2023-11-21
> > **Thanks for your response**
> >
> > Thanks for your response. The writing has been improved, and thus I raise the score of presentation to 3 (good). I have some additional comments.
> >
> > The general GNTK formula beyond two layers is given in appendix B.2 [1]. While I understand yours is more general with various GNNs and consideration of weight/bias variances, I still think differentiating what is new in this paper and what has been studied in previous works is essential. The current introduction and Section 4.1 imply that both providing GNTK and GNNGP for the simplest case have not been done before. Given GNNGP has also been comprehensively studied in [3], I find current writing inappropriate.
> >
> > My another major concern is the significance of this work is not clear, though you have some updates for the theoretical insights. Exploring why these results are useful can also greatly improve the paper.
> >
> > [3] Graph neural network-inspired kernels for gaussian processes in semi-supervised learning, in ICLR 2023.

---

> ### Author Response · Authors · 2023-11-21
> **Adressing Response**
>
> Thanks for feedback. We updated the Introduction to make more precise our contribution.
> In the related work section (and Evaluation Section) we had already quoted [3] ("Graph neural network-inspired kernels for gaussian processes in semi-supervised learning") and emphasize the fact that they already derived the GP for Graph Neural Networks. The introduction now also says that that [3] has already derived the GP for GNN and a variant of Skip-GNN.
> We updated Related Work Section, quoting the mentioned paper [1] and going more into detail.
>
> We added a section in the Introduction on the theoretical implications, later elaborated.
> And we already mention how the theoretical implications can be useful (after the proof of GNTK) and Discussion Section.
> We mention how many papers could prove theoretical properties of Neural Nets (for example Generalization Bound, Memorization Properties and Achieving Zero Training Loss for Overparametrized Neural Nets) and how our formula can
> be used to generalize these results for GNNs and also explain how the graph adjacency matrix and it's scaling is related to some of these properties (psd of Kernel).
>
> The general GNTK formula of B.2 [1] is from "Graph Neural Tangent Kernel: Fusing Graph Neural Networks with Graph Kernels" (See BLOCK Operation) which we addressed in detail also in the related work section. The problem with that formula is that it is true that it can be used for generic number of layers, but if one proceeds to use the framework to actually compute the concrete expression for a two layer GNTK one has to perform all the steps from B.2 (1,5 pages of computation) to end up with Lemma 2 (the final expression for a two layer GNTK, which is much more complicated than ours). It is very cumbersome to extend this approach to more than two layers (not to mention different architectures). None of these steps, have to be done with our formula as it is recursive.
>
> Also note that Equation 24 is exactly the formula for the **NTK**. The final equation for the GNTK is exactly the same as the NTK because all the Graph Information is encoded implicitly. We derived $\Theta$ completely differently (based on properties of Kronecker Product, vec() and Chain Rule with Jacobians).
> This way we ended up with further simplifications (than B.1 or B.2) and only as a result of our final expression (which has the adjacency matrix explicit) do certain statements, like: GNTK generalizes NTK or psd. of Kernel based on adjacency normalization pop out of the closed form. Statements about the spectrum of the Kernel are not possible with closed form from [1].
>
> **Summarizing:**
>
> previous works has touched on the GNTK on Node Classification/ Regression primarily because it is a fundamental building block for GNTK for Graph Classification/ Regression.
> Previous work failed to derive the formula in it's most general form ending up with closed forms which are significantly more complicated and do not generlize to arbitrary layers (in the same way that our formula does). Our formula is derived completely differently and can simplify extending formulas ending up in a very simple and concise closed form expression. Using our expression we can make connection on how GNTK generlizes the NTK and the Spectrum of the Kernel (how different adjacency matrices can lead to the Kernel being psd). Additionally opening further research directions which are generalizing work done on Neural Nets and are related to the spectrum of the GNTK.
> All the previous work is not concerned primarily with an in depth analysis for the GNTK on Node Classification /Regression. [1] uses it only to prove some statements later on. No previous work has any evaluations or any simulation on NTK Theory (as it wasn't their main concern, maybe because their closed form are complicted to implement).
>
> It is  important to emphasize the total efforts made in this work to delve deeply into the GNNGP and GNTK for Node Classification/Regression:
> - Deriving the GNTK and GP (for three architectures) in their most general form (the GP for vanilla GNNs and GP skip GNN is not new)
> - Identifying Weaknesses/ Errors with previous work on deriving the GNTK for Node Regression/ Classification (by showing how GNNs generalize NN and how the GNTK generalizes the NTK)
> - Evaluating the GNTK (and all the other architectures) on 7 datasets (the evaluation for the GP for vanilla GNNs and GP for a variant of the Skip-GNN has been already done)
> - Empirical Analysis on the crucial Role of $\sigma_{b}^{2}$ for Regression
> - Applying Effective Resistance for GNTK and GNN as a Graph Sparsification Method
> - Simulations and explanation how graph normalization affects wether the GNTK obeys NTK Theory.
> - Runtime Analysis, etc.
>
> For anyone that wants to proceed to further investigate properties like the spectrum of the GNTK or GNTK Theory in general will benefit from our derivations, evaluations, implementation and in depth analysis of the GNTK and GP (for various architectures).

---

> ### Comment · Reviewer_UHJu · 2023-11-22
>
> Thanks for the timely response. I think the paper has been improved a lot, and thus I raise the score to 5. I have some additionally comments.
>
> I think equation(23) and equation(24) in [1] already provide recursive formula for node GNTK, the difference with yours is just whether it is written in the vector form or the matrix form, and some reorganization of equations. I do not think you should claim theirs is not recursive or it is equivalent to NTK, which is obviously incorrect.
>
> For [3], I think you should give credit to this paper in section 3.1 given that you theorem 3.1 has been proven in theirs (regardless of minor differences), just like how you write theorem 2.1. In introduction, it should be mentioned together with other works on "extends infinite width NN theory to GNN" (such as [1] and [4]) as a background introduction of what previous works have done in this direction. Otherwise your statement of contribution might be an overclaim.
>
> Overall, this paper has good contributions though I still find many parts are informal or imprecise and can be further improved.
>
> [4] Graph Neural Tangent Kernel: Fusing Graph Neural Networks with Graph Kernels

---

> ### Author Response · Authors · 2023-11-22
> **Thanks for the Response.**
>
> Thanks again for the feedback. We made small changes to the related work section (and Theorem 3.2, on psd of GNTK). Mainly removing the claim that it is equivalent to NTK formula. We added a citation of [3] in Theorem 3.1 mentioning that [3] had already derived the GNNGP.  We quoted [4] in the introduction.
>
> Going from the vector form to the matrix form is (in our opinion) not straightforward only in hindsight once you have the matrix form. Yes the formula is recursive but the authors [1] proceeded to give an actual expression (for the two layer case) therefore they ended up with all calculations necessary for Lemma 2 (B.2), approximately (1,5) pages. For our case the formula does not require any calculations beyond the actual recursive closed form expression that we derived for any number of layers.
>
> Thanks again for the discussion.

---

> > ### Comment · Reviewer_UHJu · 2023-11-22
> >
> > Thanks for your reply. I appreciate the improvements made so far, and raised the score to 6. I would suggest that the authors carefully check the formality of the paper (especially in terms of writing) if you want to further improve it in future versions.

---

> > > ### Author Response · Authors · 2023-11-22
> > > **Thanks**
> > >
> > > Thanks for raising the score and the effort/remarks/discussion which helped to significantly improve the paper. We will put more emphasis on the formality of the writing in the future.

---

### Official Review · Reviewer_V34A · 2023-10-28

**Soundness:** 2 fair
**Presentation:** 3 good
**Contribution:** 3 good
**Rating:** 6
**Confidence:** 4

**Summary:**

In principle, the main contribution of this paper is to revisit the theory on expressive function space generated by several typical graph neural network architectures when the width of layer features goes infinity. The research question was partially answered in the previous research where the answers are not entirely correct.

**Strengths:**

1. Identify the issue in the previous literature such as Sabanayagam et al 2022 and show its incorrectness by looking at the special cases (without graph structure).

2. Clearly provide and prove the new conclusion on the graph neural tangent kernels for a number of widely used graph neural network architectures.

**Weaknesses:**

Although the main conclusions are correct (based on my reading), the paper does show certain weakness in its current format. For example, the notation used sometimes is quite confusing.  Here is an example, in the two lines before equation (1), the authors use $X\in \mathbb{R}^{d_0\times n}$ and $X\in \mathbb{R}^{n\times d_0}$, also a typo in (1) where shapes do not match with each other.    I also note that the paper on this page of openreview is different from that in pdf version.

The paper has been greatly improved.  I believe it satisfies the publication standard.

**Questions:**

Nil

---

> ### Author Response · Authors · 2023-11-15
> **Adressing Review**
>
> Thanks for the review. We fixed most notational issues. We significantly improved the paper and would hope for the reviewer to reread the new version and reconsider his/her evaluation. We gave an in depth rebuttal to many weakness as a comment to the review of Reviewer UHJu below and we would hope for the reviewer to also read it. Thanks!
> The arvix version is now the same as here just with a table of content in the beginning.
>
> TLDR:
>
> The GNTK formula in its generality, including Simulations on NTK Theory, inlcuding extensive evaluation is new and has not been done before. Theoretical Insights are added to the paper (mainly about the psd property of the Kernel w.r.t. different Graph Scalings).
> The formula can help understanding different Adjacency Matrices and their effects to for example achieving zero training loss.
> Formula and derivations for GAT*NTK/GP and Skip-GNTK/GP are also new. We added in the discussion section why we think it's important to compare different Architectures. Also small section on Effective Resistance (Spectral Graph Sparsification) on Graph Kernel methods and small section on the important Role of $\sigma_{b}^{2}$ for Regression is novel and interesting.
>
> We added to the paper:
> - New dataset: Wiki + Experiments
> - Different Nonlinearities for GAT Experiments
> - Experiment on Different Widths for GNNs
> - Runtime Analysis & Memory Analysis
> - Discussion Section
> - Theoretical Insights (psd of Kernel Matrix after GNTK proof)
> - Fixed Bug on GAT*NTK/GP

---

> ### Comment · Reviewer_V34A · 2023-11-22
> **Paper title**
>
> I meant the paper title on this submission is different from the title in pdf version.
>
> Thanks for updating the paper.

---

> > ### Author Response · Authors · 2023-11-22
> > **Changed Title**
> >
> > Thanks for the remark. We changed the pdf title to: GRAPH-NEURAL TANGENT KERNEL/GAUSSIAN PROCESS
> > FOR NODE CLASSIFICATION/REGRESSION. We would have prefered to change the openreview title but that is not allowed. There is still a minor difference with hyphenation to improve readability and conciseness.

---

> > > ### Comment · Reviewer_V34A · 2023-11-22
> > >
> > > Thanks for revising the paper.  A great improvement. Will increase my rating.

---

### Official Review · Reviewer_s1WM · 2023-11-02

**Soundness:** 3 good
**Presentation:** 3 good
**Contribution:** 3 good
**Rating:** 6
**Confidence:** 3

**Summary:**

The paper builds Kernels (neural tangent kernels) and Gaussian Processes for graph neural networks in the regime of infinite width networks. The Kernel and Gaussian Process closed forms are derived for a variety of architectures, namely the standard Graph Neural Network, the Graph Neural Network with Skip-Concatenate Connections and the Graph Attention Neural Network. The proposed methods are evaluated on transductive node classification and regression.

**Strengths:**

The paper builds on prior work on the analysis of graph neural networks in the infinite width regime and extends the current theory to the Skip-Concatenate Graph Neural Network and the Graph Attention Neural Network. Close form solutions are developed, which is a step forward to the development of the underlying theory.

The derived Gaussian Processes (GNNGP) and the Kernels (GNTK) are evaluated on six node classification/regression datasets. The results show that the new models perform better than the previous models that do not consider infinite width.

**Weaknesses:**

The new theory is with respect to GAT and skip-concatenate models. As the experimental results are showing, they are not the best models. And the theory for the nest model, GCN, is already developed. This lessens the impact of the work.

I would have liked to see an analysis of the results. Why is GNTK consistently doing better than the other models, especially with respect to GNN?

I would have liked to see the experimental results on the kernels developed by others (Du et al and Niu et al).

**Questions:**

Please see the weaknesses.

---

> ### Author Response · Authors · 2023-11-15
> **Addressing Reviews**
>
> We argue that the theory for GCN is not developed. The GCN NTK formulas in their most general form are first derived here. The GNTK is first evaluated in this work and simulations on GNTK are first ever done in this work. Du et al only considers Inductive Graph Learning Tasks. As of now we do not know why a particular models performs better than another model.
> We gave an in depth rebuttal to many weakness as a comment to the review of Reviewer UHJu below and we would hope for the reviewer to also read it. The paper has been significantly improved by incorporating the reviews. Thanks !
>
> TLDR:
>
> The GNTK formula in its generality, including Simulations on NTK Theory, inlcuding extensive evaluation is new and has not been done before. Theoretical Insights are added to the paper (mainly about the psd property of the Kernel w.r.t. different Graph Scalings).
> The formula can help understanding different Adjacency Matrices and their effects to for example achieving zero training loss.
> Formula and derivations for GAT*NTK/GP and Skip-GNTK/GP are also new. We added in the discussion section why we think it's important to compare different Architectures. Also small section on Effective Resistance (Spectral Graph Sparsification) on Graph Kernel methods and small section on the important Role of $\sigma_{b}^{2}$ for Regression is novel and interesting.
>
> TLDR:
> The GNTK formula in its generality, including Simulations on NTK Theory, inlcuding extensive evaluation is new and has not been done before. Theoretical Insights are added to the paper (mainly about the psd property of the Kernel w.r.t. different Graph Scalings). The formula can help understanding different Adjacency Matrices and their effects to for example achieving zero training loss. Formula and derivations for GAT*NTK/GP and Skip-GNTK/GP are also new. We added in the discussion section why we think it's important to compare different Architectures. Also small section on Effective Resistance (Spectral Graph Sparsification) on Graph Kernel methods and small section on the important Role of for Regression is novel and interesting.
>
> We added to the paper:
> - New dataset: Wiki + Experiments
> - Different Nonlinearities for GAT Experiments
> - Experiment on Different Widths for GNNs
> - Runtime Analysis & Memory Analysis
> - Discussion Section
> - Theoretical Insights (psd of Kernel Matrix after GNTK proof)
> - Fixed Bug on GAT*NTK/GP

---

> > ### Comment · Reviewer_s1WM · 2023-11-22
> > **Thanks for the response.**
> >
> > Thanks for the response. I will maintain my score as I consider that to be consistent with the significance of the work.

---

### Meta-Review · Area_Chair_Xvr3 · 2023-12-14

**Metareview:**

This paper derives closed-form expressions for graph kernels in the infinite width limit of graph neural networks, for node classification and regression, for a number of common architectures. The results expand upon existing results, in some cases fixing issues in existing derivations, in others providing more natural expressions and extensions to arbitrary numbers of layers, and in others (e.g. for graph attention networks) where no closed-form expressions had been previously published.

Overall based on the reviews, the paper is borderline, though reviewers did raise their scores following discussion with the authors and further refinement of the paper. At this point, the main issues are related to presentation and interpretation, as well as the general significance. Results and discussions throughout the paper are generally presented in an informal and/or imprecise way, e.g. experimental results with no standard deviation, which has not been addressed in rebuttal. In addition to that, the significance of these results are still unclear; though the paper mentions how these results can be potentially used by future works, the paper itself does not really address any unanswered theoretical questions or offer novel insights for understanding GNNs more broadly. The computation of the kernel is also not practically scalable. There are also minor issues with the writing, for example abuse of capitalization.

In my opinion, while this is technically correct and valuable work, and it did improve quite a lot during the rebuttal phase, I think it is not quite ready for ICLR in its current form and would benefit from a fresh round of review on the revision. The authors should focus on a clear presentation of their theorems, and a clear delineation of their contributions relative to existing work, in the revised version.

**Justification For Why Not Higher Score:**

All reviewers had issues with presentation, and with significance of the contribution relative to related work. While these were addressed quite a bit by the revision, in private discussion between the reviewers there was agreement that this was still borderline and probably just under the bar for acceptance still. However, it would be fine to accepting it, as the major issues have been resolved.

**Justification For Why Not Lower Score:**

N/A

---

### Decision · Program_Chairs · 2024-01-16

Reject